# Differential modulation of polycomb-associated histone marks by cBAF, pBAF, and gBAF complexes

Mary Bergwell[1] , JinYoung Park[1,2] , Jacob G Kirkland[1,2]

Chromatin regulators alter the physical properties of chromatin to make it more or less permissive to transcription by modulating another protein's access to a specific DNA sequence through changes in nucleosome occupancy or histone modifications at a particular locus. Mammalian SWI/SNF complexes are a group of ATPase-dependent chromatin remodelers. In mouse embryonic stem cells, there are three primary forms of mSWI/SNF: canonical BAF (cBAF), polybromo-associated BAF (pBAF), and GLTSCR-associated BAF (gBAF). *Nkx2-9* is bivalent, meaning nucleosomes at the locus have active and repressive modifications. In this study, we used unique BAF subunits to recruit each of the three complexes to *Nkx2-9* using dCas9-mediated inducible recruitment (FIRE-Cas9). We show that recruitment of cBAF complexes leads to a significant loss of the polycomb repressive-2 H3K27me3 histone mark and polycomb repressive-1 and repressive-2 complex proteins, whereas gBAF and pBAF do not. Moreover, nucleosome occupancy alone cannot explain the loss of these marks. Our results demonstrate that cBAF has a unique role in the direct opposition of polycomb-associated histone modifications that gBAF and pBAF do not share.

## Introduction

The eukaryotic genome is organized into three primary chromatin states: accessible euchromatin, conditionally accessible facultative heterochromatin, and mostly inaccessible constitutive heterochromatin. DNA accessibility is essential for regulating transcription factor binding; therefore, these accessibility states contribute to the transcriptional state, with the most accessible euchromatin being largely transcriptionally active and the two heterochromatin states being transcriptionally repressed. Facultative heterochromatin is generally defined by polycomb group (PcG) proteins and their respective histone marks. Polycomb consists of two main repressive complexes, polycomb repressive complex 1 (PRC1) and polycomb repressive complex 2 (PRC2), that modify histones as a part of transcriptional silencing. PRC1 writes and reads the H2AK119ub modification, whereas PRC2 writes and reads the H3K27me3 histone modification (1). There is an additional cross-talk between these two complexes. Regulation of facultative heterochromatin—polycomb proteins and their respective histone modifications—is critical during development. For a cell to differentiate, it must coordinate new transcriptional programs (2, 3, 4). At each point in the cell-fate decision-making tree, new sets of genes must be activated, whereas other sets of genes must be repressed. The mSWI/SNF family of protein complexes (also known as BAF complexes (5)) is chromatin proteins that help regulate the balance of facultative heterochromatin in the cell and are required to modulate transcriptional states during development (5, 6, 7, 8, 9, 10). In stem cells, there are three distinct, major forms of BAF complexes often in the same cell: canonical BAF (cBAF), polybromo-associated BAF (pBAF), and GLTSCR-associated BAF (gBAF) (11, 12, 13). The three forms share common subunits, but each complex has distinct subunits not shared by the others.

Although the earliest SWI/SNF studies described its role in nucleosome remodeling (14), subsequent studies revealed its role in regulating chromatin accessibility and transcription through opposition to polycomb group proteins (7, 8, 9, 10). To understand how BAF complexes evict polycomb and activate genes, we developed a method of rapamycin-induced chromatin regulator recruitment to specific loci called FIRE-Cas9 (15). FIRE-Cas9 uses catalytically dead Cas9 and sgRNAs to target a particular locus. Frb and Fkbp fusion proteins are then used to force chemical-induced proximity at the dCas9-bound locus by adding rapamycin to the cell culture media. The FIRE-Cas9 system enables the recruitment of a specific chromatin regulator and the tracking of the resulting consequences with minute-by-minute kinetics on physiologic chromatin (15). Our previous study used the BAF subunit SS18 as the recruitment anchor (15). Although it was not known then, SS18 is present in both cBAF and gBAF complexes (12, 13, 16). We found that recruiting SS18-containing complexes led to the rapid eviction of polycomb proteins, and this preceded transcriptional activation, thus elucidating the order of events of transcriptional regulation on the chromatin level by BAF complexes (8, 15). However, because

[1]Cell Cycle and Cancer Biology Program, Oklahoma Medical Research Foundation, Oklahoma City, OK, USA    [2]Department of Cell Biology, University of Oklahoma Health Science Center, Oklahoma City, OK, USA

Correspondence: jake-kirkland@omrf.org
Mary Bergwell's present address is Genetics and Epigenetics Program, University of Pennsylvania, Philadelphia, PA, USA

both cBAF and gBAF share the SS18 subunit, it is unclear whether one or both complexes oppose the polycomb repressive marks. The ability of the third major complex, pBAF, to do the same also remains unknown.

In this study, we describe a modification of the FIRE-Cas9 system using tagged subunits that are found in only one of the three complexes as recruitment anchors. By integrating the unique subunits DPF2 (cBAF), BRD9 (gBAF), or PHF10 (pBAF) into our FIRE-Cas9 system, we can specifically recruit cBAF, gBAF, or pBAF. With this approach, we can perform experiments that will define and distinguish the contributions of the three complexes to polycomb-associated histone modifications at the bivalent *Nkx2-9* gene. The recruitment studies presented here show differential modulation of polycomb-associated histone marks by cBAF, gBAF, and pBAF complexes.

## Results

### Tagged unique BAF subunits are expressed and incorporated into BAF complexes

To achieve better fusion protein expression, we modified the FIRE-Cas9 system by switching the Frb/Fkbp dimerization tags. Previously, the Fkbp tag was attached to the MS2 bacteriophage coat protein (MS2), and the Frb tag was attached to our chromatin regulator of interest. In this study, we have now tagged MS2 with Frb (MS2-2xFrb) and our chromatin regulator with Fkbp domains (CR-2xFkbp-V5) (Fig 1A). We used lentiviral transduction to express each of the FIRE-Cas9 components in mouse embryonic stem cells (mESCs) (Fig 1B). Our experimental lines were built by first transducing with expression plasmids for dCas9-HA and MS2-2xFrb, ensuring equal expression in the final lines (Fig 1C). We then transduced cells with DPF2-2xFkbp-V5, BRD9-2xFkbp-V5, or PHF10-2xFkbp-V5 (Fig 1D). PHF10 has multiple isoforms. This study uses an isoform containing a c-terminal PHD or DPF domain (now known as PHF10-P) which was the originally described form (17). The PHD/DPF domain has been shown to be required for transcriptional activation (18, 19). We finally transduced these lines with three sgRNAs targeting the promoter of *Nkx2-9* that also contains extra stem-loop structures to complete the system. The sgRNAs guide the dCas9 to the *Nkx2-9* promoter, and the MS2-Frb protein binds the stem-loops. Only after adding rapamycin are specific BAF complexes recruited to *Nkx2-9* by the dimerization of the Frb and Fkbp domains (Fig 1A). dCas9 is always present at the locus in our control (no rapamycin) and experimental conditions (rapamycin), allowing for better comparisons than methods where a chromatin regulator is directly tethered to dCas9. A key advantage of chemical-induced proximity (20) is that the rapamycin selectively binds to both Frb and Fkbp at once with an affinity 2,000-fold tighter than binding to the Frb domain (found in mTOR) alone, which allows us to use rapamycin at low doses (3 nM) in FIRE-Cas9 experiments (21). We performed a series of co-immunoprecipitations to ensure that our tagged proteins were incorporated into BAF complexes. First, we immunoprecipitated with a mouse anti-V5 antibody to pull down our CR-2xFkbp-V5 fusion proteins and then performed Western blots to other shared BAF subunits BAF155 (SMARCC1) and BRG1

(SMARCA4). BAF155 forms a dimer as the initial BAF core in the first step of BAF complex assembly (22). Only after this core is formed does the assembly process diverge for the three major complexes (22). The unique subunits tagged in this study are subsequently added, creating the cBAF, gBAF, and pBAF cores (22). Finally, the ATPase module, which includes BRG1, is added in one of the final steps of complex assembly (22). Here, we show that pulldown using an antibody to the V5 epitope, DPF2-2xFkbp-V5, BRD9-2xFkbp-V5, and PHF10-2xFkbp-V5 all interact with BAF155 and BRG1, suggesting they are part of full BAF complexes (Fig 1E). A WT line that does not have a V5 epitope fails to pulldown BAF155 and BRG1 (Fig 1F). To further confirm PHF10-2xFkbp-V5 interactions with pBAF-specific subunits, we performed a pulldown with ARID2. These data show that ARID2 interacts with the tagged PHF10-2xFkbp-V5 subunit (detected with a V5 antibody), along with BAF155 and BRG1 (Fig 1G). Detection of PHF10-2xFkbp-V5 was specific to our expressing line and was not found in WT TC1 cells. These data suggest that DPF2-2xFkbp-V5, BRD9-2xFkbp-V5, and PHF10-2xFkbp-V5 are all incorporated into their respective BAF complexes.

### Recruitment of specific BAF complexes via unique subunits

After showing that our tagged DPF2, BRD9, and PHF10 subunits are incorporated into complete BAF complexes, we recruited cBAF, gBAF, or pBAF to the *Nkx2-9* promoter. Here, we added rapamycin for 24 h to achieve prolonged chromatin regulation by BAF complexes, followed by fixation and chromatin immunoprecipitation (ChIP) and qPCR to evaluate recruitment-based changes in polycomb-associated histone marks. To show that we are recruiting our tagged subunits to the *Nkx2-9* promoter, we performed ChIPs using a V5 antibody because each of our recruitment subunits contains this epitope, followed by qPCR (Fig 2A and B). DPF2-based cBAF and BRD9-based gBAF recruitment showed significant enrichment at the −315 and +160 amplicons near the recruitment site (Fig 2A and B). cBAF has a higher median enrichment than gBAF, but these two datasets have overlapping data points and are not statistically significant from each other (Fig S1A). However, PHF10-based pBAF recruitment failed to show enrichment via the V5 antibody (Fig S1B). We then performed ChIP against the PHF10 subunit itself and found robust enrichment of the recruited subunit (Figs 2C and S1C). This observation complicates our ability to directly compare the overall recruitment levels of the different complexes because even the same antibody shows a differential ability to bind the epitope (V5 in this case) depending on the subunit tagged and the complex it incorporates into. However, in each case, enrichment of the recruited subunit is specifically around the sgRNA binding sites (spanning −172 to −25 bp) and does not spread far up or downstream.

### Only cBAF complexes robustly oppose polycomb-associated histone marks

Next, we evaluated polycomb-associated histone marks. H3K27me3 is associated with PRC2 complexes, whereas H2AK119ub is associated with PRC1 complexes (1). Here, we found critical differences between the BAF complexes. Our previous work showed that SS18-containing complexes led to the loss of repressive H3K27me3

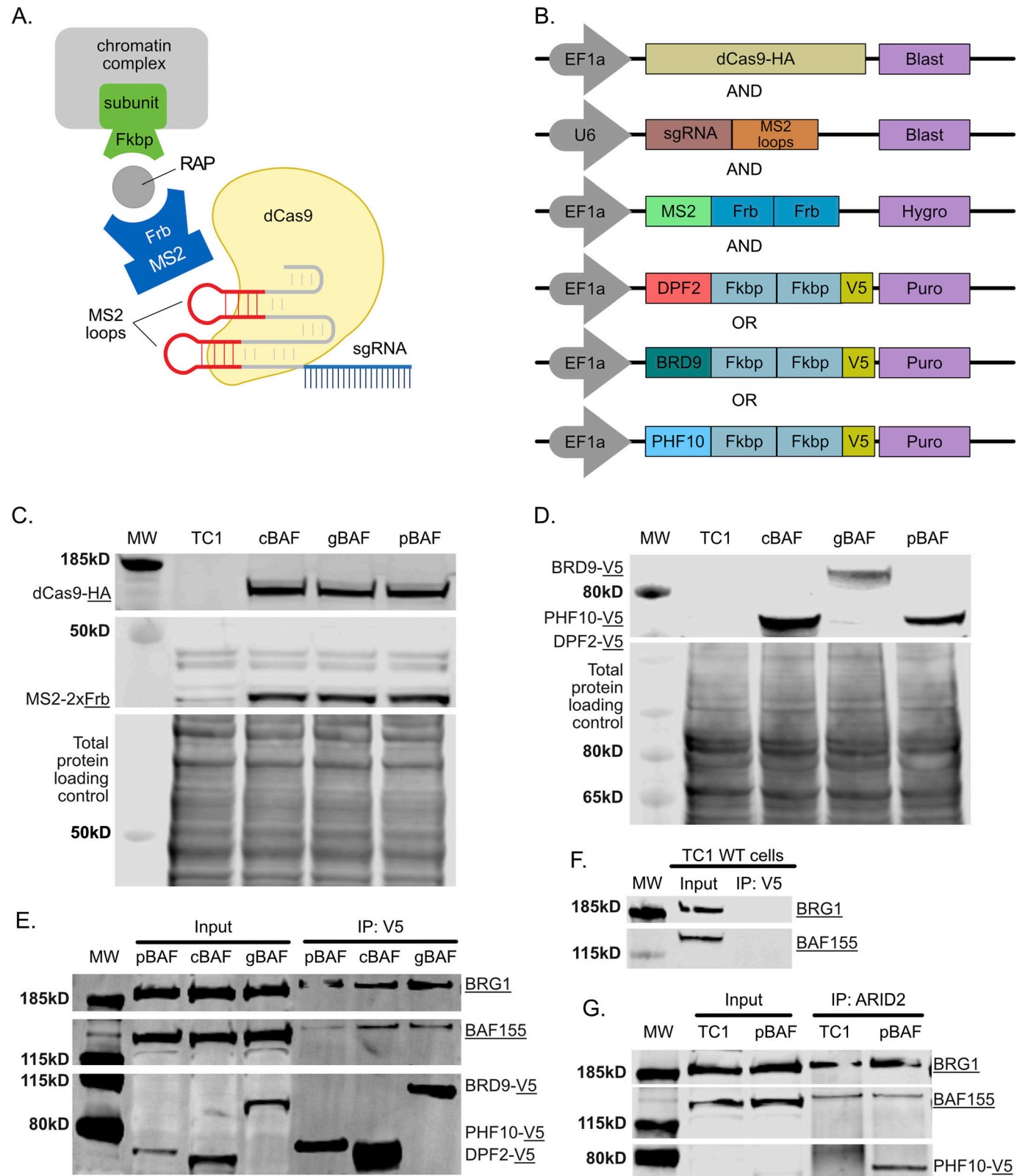

**Figure 1. Tagged unique BAF subunits are expressed and incorporated into BAF complexes.**
**(A)** Schematic of the modified FIRE-Cas9 system showing induced proximity by dimerization of Frb and Fkbp domains by rapamycin. **(B)** Lentiviral expression constructs of FIRE-Cas9 components. **(C)** Western blot showing expression of dCas9-HA and MS2-2xFrb in each cell line with WT TC1 negative and total protein staining controls. **(D)** Western blot showing expression of 2xFkbp-V5–tagged BAF subunits used to recruit their respective complexes with WT TC1 negative and total protein staining controls.

histones upon recruitment to *Nkx2-9* (15). However, because both cBAF and gBAF contain SS18, it remained unclear whether both or only one of these complexes was responsible for this observation. We now show that cBAF complexes are predominantly responsible for H3K27me3 loss with a minor but not statistically significant gBAF-mediated loss (Fig 3A and B). Like gBAF, pBAF complexes only provide minimal loss of the H3K27me3 mark and only directly at the recruitment site, failing to spread like cBAF-mediated recruitment (Fig 3C). All three complexes evicted H3K27me3 better than H2AK119ub, including cBAF complexes (Fig 3D–F). Once again, cBAF complexes were the most robust evictor of PRC1-associated marks (Fig 3D). In addition, eviction of H2AK119ub occurred most strongly at the recruitment site and spread upstream of the promoter but not downstream of the promoter in the gene body in contrast with H3K27me3 (Fig 3D–F). Only two sites showed a statistically significant drop after the recruitment of cBAF: the recruitment site and the distal upstream site (−315 and −2,455 bp). Because of the discrepancy between PRC1 and PRC2-associated histone marks and others showing that H2AK119ub ChIPs have high background levels (23), we performed ChIPs to the PRC2 protein SUZ12 and the PRC1 protein RING1B to attempt to resolve how cBAF-mediated opposition of polycomb may function. We had not assayed for H2AK119ub in our previous SS18-based recruitment study (15). Here, we show a loss of SUZ12 along the entire *Nkx2-9* locus, spreading further than just the recruitment site, similar to H3K27me3 (Fig 3G).

In contrast to the PRC1-associated histone mark H2AK119ub, we saw a significant loss of the PRC1 protein RING1B itself along the *Nkx2-9* locus, spreading upstream further than it did downstream (Fig 3H). The last +1,400 amplicon was down but not significantly changed. Together these data suggest that cBAF effectively evicts both PRC1 and PRC2 complexes and the PRC2-associated histone mark H3K27me3. Finally, we were concerned that recruitment efficiency was directly correlated with the loss of H3K27me3, so we compared V5 enrichment and H3K27me3 enrichment for each biological replicate and performed a linear regression analysis (Fig S2). If the most highly recruited replicates led to the most robust H3K27me3 eviction, we would have expected a negative correlation between V5 and H3K27me3 enrichment. However, we had no correlation or slightly positive correlations suggesting that the strength of recruitment is not directly correlated with the strength of PRC-associated histone eviction. Finally, we were able to show significant enrichment of other BAF subunits for each of the three types of BAF complexes after recruiting each BAF complex (ARID1A, cBAF; SS18, gBAF; BAF57, pBAF; Fig S3).

### Nucleosome depletion is insufficient to explain cBAF-mediated loss of polycomb-associated histone marks

Loss of a histone mark can occur through several mechanisms: (1) co-recruitment of an enzyme or enzymes that remove the marks

(demethylases and deubiquitinases in this case), leaving the rest of the nucleosomes intact; (2) removal of a nucleosome, leaving a nucleosome-depleted region (NDR); or (3) removal of a nucleosome which is then replaced with an unmodified nucleosome. To test these possibilities, we performed recruitment experiments of the three BAF complexes followed by ChIP using an antibody to the C-terminus of H3, which detects nucleosomes regardless of H3 modifications (Fig 4A–C) or Histone H4 (Fig 4D–F). Although there was some minor nucleosome loss after cBAF recruitment (Fig 4A and D), the losses were not statistically significant. Furthermore, the loss of modified histones was larger than that of all nucleosomes, suggesting that model (2) removal of a nucleosome, leaving an NDR, is insufficient to explain the loss of H3K27me3 (Fig S3A). In contrast, the changes in H2AK119ub can be explained by a potential slight loss of nucleosomes (Fig S3D). H2AK119ub normalized to H3 nucleosome occupancy showed a more peculiar pattern where loss upstream of the marked histone was slightly greater than that of all nucleosomes. H2AK119ub downstream of the transcription starting site (TSS) showed the opposite pattern, with a slight increase in the amount of H2AK119ub per nucleosome. Whereas none of these changes were statistically significant, the pattern may suggest differences in H2AK119ub dynamics in relation to the direction of a gene. Recruitment of gBAF led to minimal to no nucleosome depletion (Fig 4B and E), and any loss of H3K27me3 or H2AK119ub can be explained by nucleosome depletion (Fig S4B and E). Finally, pBAF recruitment led to minimal to no loss of H4 or H3 (Fig 4C and F), and the ratios of H3K27me3 and H2AK119ub on a per nucleosome level remained unchanged (Fig S3C and F).

### cBAF recruitment leads to a gain of H3.3-containing nucleosomes

Because we found that nucleosome depletion was insufficient to explain the loss of polycomb-associated histone marks, we set out to test model (3), whereby the removal of a polycomb-associated nucleosome is then replaced with an unmodified nucleosome. Outside the centromere, mammalian nucleosomes may contain one of three Histone 3 variants: H3.1/H3.2 (replication-dependent deposition) or H3.3 (replication-independent deposition). Here, we hypothesized that the replacement nucleosomes may be deposited independently of replication. H3.3 differs from H3.1 and H3.2 by 4 or 5 amino acids, so we used an antibody raised against this variable region and is specific to the H3.3 variant. Here, we found an increase in H3.3 histones after the recruitment of cBAF (Fig 5A). There was no significant gain in H3.3 after the recruitment of gBAF or pBAF complexes (Fig 5B and C). These data contrast the data when we used an antibody that recognizes all of H3.1, H3.2, and H3.3, whereby levels of H3 were slightly depleted or remained unchanged (Fig 4). The possibility remained that H3.3 levels at *Nkx2-9* were so small that a slight increase could lead to a large fold change upon recruitment of BAF complexes. To address this,

---

**(E)** Co-IP pulldown with an antibody against V5 followed by Western blot showing interactions between V5-tagged BAF subunits and the core subunit BAF155 and the ATPase BRG1. **(F)** Co-IP pulldown with an antibody against V5 followed by Western blot in a WT cell line not expressing the V5 tag showing the lack of interactions between V5-tagged BAF subunits and the core subunit BAF155 and the ATPase BRG1. **(G)** Co-IP pulldown with an antibody against the pBAF-specific subunit ARID2 showing interactions with BRG1, BAF155, and PHF10-2xFkbp-V5.
Source data are available for this figure.

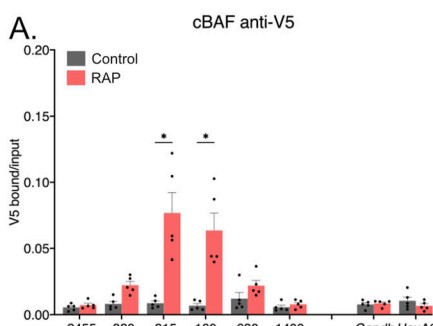
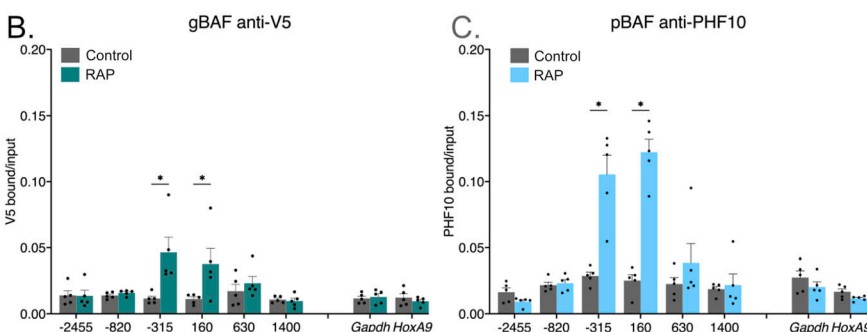

**Figure 2. Recruitment of specific BAF complexes via unique subunits.**
Chromatin immunoprecipitations followed by qPCR detecting enrichment of the unique recruited subunits. 0 bp denotes the TSS of *Nkx2-9* and the recruitment site spans −172 to −25 bp (upstream of the TSS). Data are presented as bound/input either no recruitment (control) or recruitment (RAP). **(A)** Pulldown of DPF2-2xFkbp-V5 with an antibody against V5 (cBAF). **(B)** Pulldown of BRD9-2xFkbp-V5 with an antibody against V5 (gBAF). **(C)** Pulldown of PHF10-2xFkbp-V5 with an antibody against PHF10 (pBAF) (n = 5; mean ± s.e.m.). Significance was evaluated by multiple Mann–Whitney tests. * denotes q < 0.05.

we performed CUT&TAG genome-wide on H3.3 and mapped peaks in two biological replicates (Fig S4G). Here, we found that there were indeed three reproducible H3.3 peaks within the *Nkx2-9* locus, ranging from 18 to 48 percentiles of all peaks in the genome. In addition, the H3.3 qPCR enrichment levels were similar to those at the *Gapdh* and *HoxA9* loci (Fig 5), suggesting *Nkx2-9* is not devoid of all H3.3 before recruiting BAF complexes. Therefore, these data are consistent with a model, whereby cBAF evicts nucleosomes with polycomb-associated modifications, which are then replaced with H3.3-containing nucleosomes that largely lack H3K27me3 modifications.

### cBAF is a transcriptional activator of *Nkx2-9*

Finally, we performed RT-qPCR to evaluate the ability of the different BAF complexes to promote transcriptional activation of *Nkx2-9*. We added rapamycin to the media for 24 h to recruit the various BAF complexes. cBAF complexes acted as the most robust transcriptional activators, achieving a 2.8-fold increase in Nkx2-9 mRNA. Compared with cBAF, neither gBAF (1.69-fold) nor pBAF (1.58-fold) was as strong as transcriptional activators of *Nkx2-9*, consistent with their inability to effectively evict polycomb-associated histone marks (Fig 6A). We next asked if the eviction of H3K27me3 was sufficient for transcriptional activation. To test this, we used a selective inhibitor of the PRC2 methyltransferase EZH2 (GSK126) for up to 4 d of treatment. We performed a Western blot against H3K27me3 with pan H3 as a loading control (Fig 6B). Here, we found that H3K27me3 levels were undetectable on both day two and day four. However, H2AK119ub levels remained largely unchanged. We then assayed Nkx2-9 transcripts by RT-qPCR and found no increase in *Nkx2-9* transcription in the near complete absence of H3K27me3 in the genome (Figs 6C and S4A). Whereas we could not detect H3K27me3 by Western blot, there remained the possibility that H3K27me3 still remained at the *Nkx2-9* locus. To test this, we performed ChIP-qPCR after 4 d of EZH2 inhibition. We found a substantial decrease in H3K27me3 but not H2AK119ub at the *Nkx2-9* locus, consistent with the Western blot (Fig S4B and C). Levels of H3K27me3 at the *HoxA9* locus were largely unchanged suggesting that some loci do retain H3K27me3 even with robust EZH2 inhibition.

Because the fold change in transcripts appeared modest, we used CRISPR-A (24) as a positive control. In this case, the activating Cas9 is constitutively present at *Nkx2-9* as this system is not inducible like FIRE-Cas9, so gene activation has been happening for more than 24 h. Therefore, CRISPR-A is expected to be a more potent transcriptional activator but represents a maximum activation limit. In these experiments, we compared a line expressing CRISPR-A without targeting sgRNAs with one expressing CRISPR-A with the same sgRNAs used to recruit the FIRE-Cas9 to *Nkx2-9*. We found that cBAF activated transcription of *Nkx2-9* to a level about half as strong as the CRISPR-A positive control (Fig S4D; *P* = 0.17 Welch's *t* test).

## Discussion

We previously showed that BAF complexes that contain the SS18 subunit can evict H3K27me3 and activate transcription at the *Nkx2-9* gene. At the time, SS18 had only been described as a subunit of the BAF complex, but subsequent studies separated two SS18-containing complexes: the canonical BAF (cBAF) and GLTSCR-associated BAF (gBAF) complexes. Here, we show that we can specifically recruit cBAF, gBAF, and pBAF complexes through their unique subunits DPF2, BRD9, and PHF10 using our FIRE-Cas9 inducible recruitment system. We show that of the two SS18-containing complexes, cBAF is the more potent H3K27me3 (PRC2) and H2AK119ub (PRC1) opposing BAF complex, definitively showing a separation of the function of the two SS18-containing complexes, thus resolving ambiguity introduced by the discovery of gBAF complexes.

Loss of a histone mark can occur through several mechanisms: (1) co-recruitment of an enzyme or enzymes that remove the marks, leaving the rest of the nucleosomes intact; (2) removal of a nucleosome, leaving a NDR; or (3) removal of a nucleosome which is then replaced with an unmodified nucleosome. Our data show that the cBAF-mediated loss of these marks, particularly H3K27me3, cannot be explained by nucleosome depletion alone (model 2), as more modified histones are removed after cBAF recruitment than total histones. Whereas future work is required to distinguish

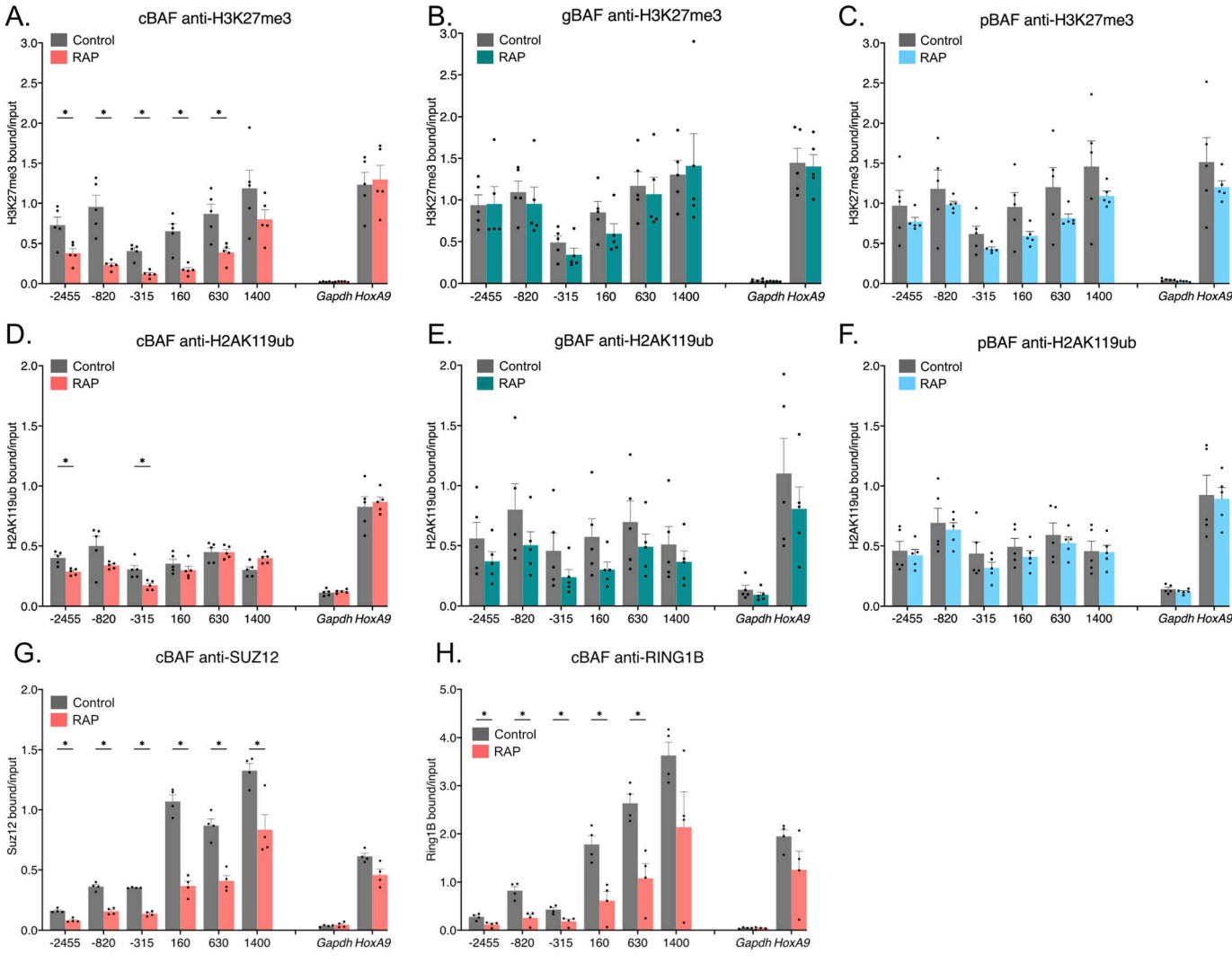

**Figure 3. Only cBAF complexes robustly oppose polycomb-associated histone marks.**
Chromatin immunoprecipitations followed by qPCR detecting enrichment of histone modifications. 0 bp denotes the TSS of *Nkx2-9* and the recruitment site spans −172 to −25 bp (upstream of the TSS). Data are presented as bound/input either no recruitment or recruitment. **(A)** Pulldown of H3K27me3 after recruitment of cBAF complexes (n = 5; mean ± s.e.m.). **(B)** Pulldown of H3K27me3 after recruitment of gBAF complexes (n = 5; mean ± s.e.m.). **(C)** Pulldown of H3K27me3 after recruitment of pBAF complexes (n = 5; mean ± s.e.m.). **(D)** Pulldown of H2AK119ub after recruitment of cBAF complexes (n = 5; mean ± s.e.m.). **(E)** Pulldown of H2AK119ub after recruitment of gBAF complexes (n = 5; mean ± s.e.m.). **(F)** Pulldown of H2AK119ub after recruitment of pBAF complexes (n = 5; mean ± s.e.m.). **(G)** Pulldown of SUZ12 after recruitment of cBAF complexes (n = 4; mean ± s.e.m.). **(H)** Pulldown of RING1B after recruitment of cBAF complexes (n = 4; mean ± s.e.m.). Significance was evaluated by multiple Mann-Whitney tests. * denotes q < 0.05.

between models 1 and 3, we currently have no data consistent with model 1. Others have shown that BAF competes with PRC complexes to evict H3K27me3 without the co-recruitment of demethylases in a hematopoietic stem cell system, consistent with model 3 (25). These data suggest that the mode of H3K27me3 opposition may be shared across loci and cell types. For model 1 to be correct, cBAF must recruit KDM6A or KDM6B (26) to remove H3K27me3 marks and BAP1/ASXL1 (27, 28) to remove H2AK119ub marks at the same sites. This co-recruitment seems unlikely compared with model 3, where cBAF removes polycomb-associated nucleosomes, thereby removing H3K27me3 and H2AK119ub simultaneously. After the removal of these nucleosomes, new nucleosomes containing H3.3 but lacking H3K27me3 are inserted into the chromatin, which cBAF complexes

may have a lower affinity for removing once again compared with polycomb-associated H3.1-containing nucleosomes. This specificity may be through interactions with PRC1 proteins themselves (8, 9), which are bound at these modified nucleosomes and are not found at the unmodified nucleosomes, thereby instructing cBAF which nucleosomes are to be preferentially evicted. However, our observation that H3K27me3 is removed more efficiently and over a broader region than H2AK119ub suggests additional undiscovered mechanisms of PRC2 eviction. cBAF may also preferentially avoid evicting H3.3-containing nucleosomes regardless of the polycomb-associated histone modifications. The BAF subunits BRG1 and SMARCB1 interact with the H3.3 chaperone HIRA complex (29). This interaction may direct or promote H3.3 deposition at BAF-regulated

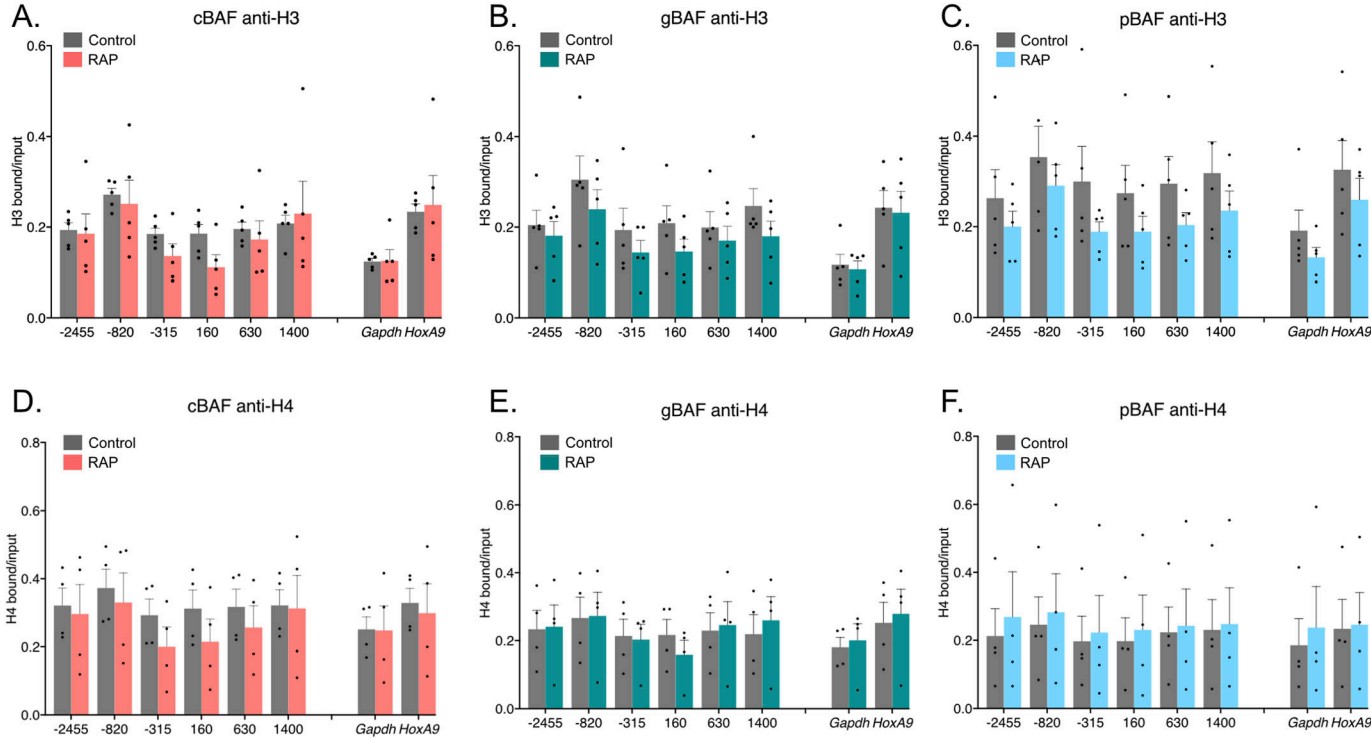

**Figure 4.  Nucleosome depletion is insufficient to explain cBAF-mediated loss of polycomb-associated histone marks.**
Chromatin immunoprecipitations followed by qPCR detecting enrichment of histones and their modifications. The recruitment site is approximately −100 bp. 0 bp denotes the TSS of *Nkx2-9*. **(A)** Pulldown of pan H3 after recruitment of cBAF complexes. **(B)** Pulldown of pan H3 after recruitment of gBAF complexes. **(C)** Pulldown of pan H3 after recruitment of pBAF complexes. **(D)** Pulldown of H4 after recruitment of cBAF complexes. **(E)** Pulldown of H4 after recruitment of gBAF complexes. **(F)** Pulldown of H4 after recruitment of pBAF complexes. Data are presented as bound/input either no recruitment or recruitment (n = 4–5; mean ± s.e.m.). Significance was evaluated by multiple Mann-Whitney tests.

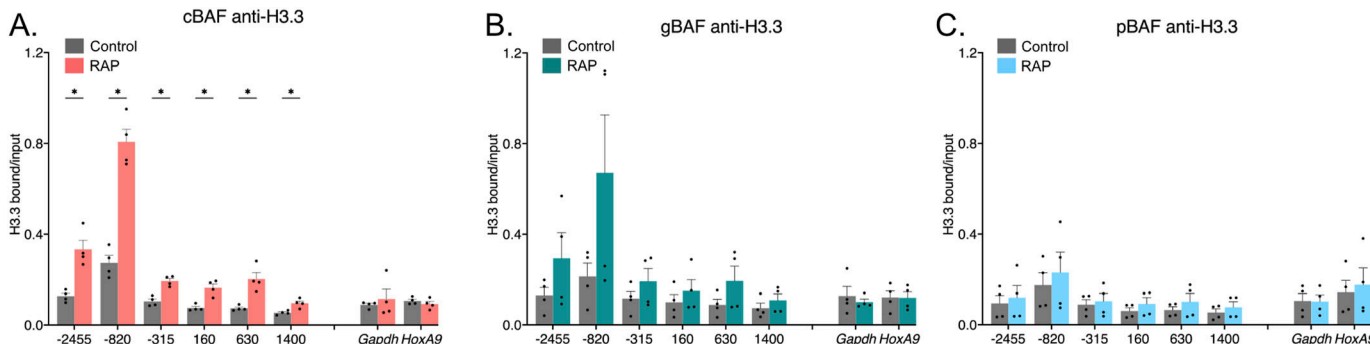

**Figure 5.  cBAF recruitment leads to a gain in this histone variant H3.3.**
Chromatin immunoprecipitations followed by qPCR detecting enrichment of histones and their modifications. The recruitment site is approximately −100 bp. 0 bp denotes the TSS of *Nkx2-9*. **(A)** Pulldown of H3.3 after recruitment of cBAF complexes. **(B)** Pulldown of H3.3 after recruitment of gBAF complexes. **(C)** Pulldown of H3.3 after recruitment of pBAF complexes. Data are presented as bound/input either no recruitment or recruitment (n = 4; mean ± s.e.m.). Significance was evaluated by multiple Mann-Whitney tests. * denotes q < 0.05.

sites, promoting transcription and a more open chromatin state (30). However, the BRG1 and SMARCB1 subunits are shared between all BAF complexes, so it remains unclear how different BAF complexes influence H3.3 deposition to varying degrees.

In addition, we found a discrepancy between the PRC1 protein (RING1B) and the PRC1-associated mark H2AK119ub removal, where the actual PRC1 proteins are evicted much more efficiently. This could be a technical artifact behind H2AK119ub ChIPs or could be biological, where cBAF complexes physically interact with cPRC1 to evict the protein complex (9, 31), but the histone mark remains more stable even without PRC1 present at the locus. An increase in PRC1-binding chromatin in BRG1 ATPase conditional knockout or mutant cells is consistent with our data showing cBAF complexes efficiently evict RING1B proteins, but the loss of H2AK119ub marks in BRG1

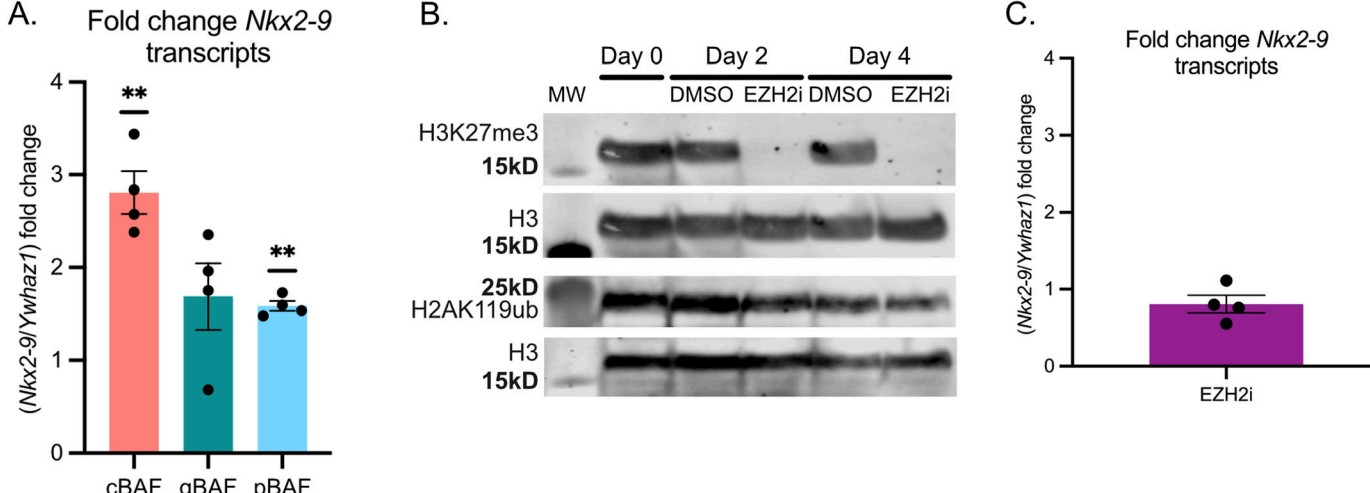

**Figure 6.   cBAF is a transcriptional activator of *Nkx2-9*.**
**(A)** RT-qPCR of the fold change of *Nkx2-9* transcription (normalized to *Ywhaz1*) after recruitment of cBAF, gBAF, and pBAF. (n = 4; mean ± s.e.m.). Significance was evaluated by a one-sample *t* test (null = 1.0) corrected for multiple samples ** denotes q < 0.01 **(B)** Western blot against H3K27me3, pan H3 (loading control), or H2AK119ub after 0, 2, or 4 d of treatment with DMSO control or EZH2 inhibitor GSK126 (3 μM). **(C)** Fold change in *Nkx2-9* transcript levels by RT-qPCR after 4 d of EZH2 inhibition compared with DMSO control (n = 4, mean ± s.e.m.). Significance was evaluated by a one-sample *t* test (null = 1.0).
Source data are available for this figure.

mutants was not explored in this study. Hence, we are unsure whether the difference between RING1B eviction and H2AK119ub is universal (9). This contrasts with PRC2 and its histone mark H3K27me3, which are both evicted with high efficiency and may speak to a difference in PRC1 versus PRC2 biology.

By inhibiting EZH2, the PRC2 methyltransferase responsible for writing the H3K27me3 mark in mESCs, we showed that simply removing H3K27me3 from most of the genome and at *Nkx2-9* gene specifically was not sufficient for transcriptional activation of *Nkx2-9*. This suggests that cBAF complex-mediated transcriptional activation is doing something other than just removing H3K27me3 from a locus to activate it. Because H2AK119ub remains at the locus and throughout the genome, it is possible that cBAF must remove both PRC1 and PRC2 proteins to activate a gene. These data are consistent with our findings and those of others that the BAF-mediated removal of PRC-associated histone marks at *Ascl1* is insufficient for transcriptional activation of the gene (8, 9, 15). Therefore, the model we favor is that cBAF complexes must remove both PRC1/2 and bring in co-factors such as P300 or transcription factors to activate a gene. These co-factors are not accessing the promoter by removing H3K27me3 alone. It remains possible that the removal of PRC1/2 and H3K27me3, but not H2AK119ub, provides modulation of transcript levels, and removing all four classes (proteins and histone marks) could lead to even more robust transcriptional activation.

We further show differential modulation of polycomb-associated histone marks by investigating pBAF complexes (which lack SS18) for the first time. Here, we show that pBAF complexes are similar to the gBAF complex in their inability to oppose polycomb-associated histone marks at *Nkx2-9* efficiently. Finally, when recruited to a bivalent gene, we show that the cBAF complex is a better transcriptional activator than the gBAF and pBAF complexes. In total, we have provided data describing different

facultative heterochromatin modulations by the three main BAF complexes in mESCs. This suggests that each BAF complex has a unique role in changing or maintaining the balance of facultative heterochromatin in the nucleus when brought to a locus through native protein-protein interactions, which may include histone modifications, transcription factors, and other chromatin regulators.

**Limitations**

One limitation of this study is that it proved impossible to measure the recruitment rates of the three BAF complexes in relation to each other. Whereas all complexes contained a V5-tagged subunit, using a V5 antibody in the PHF10-2xFkbp-V5-based pBAF recruitment experiments showed no enrichment, whereas an antibody against PHF10 did. This suggests the V5 antibody had variable accessibility to its epitope within a formaldehyde-fixed BAF complex. We hypothesize that the V5 epitope is masked within a pBAF complex by other subunits after fixation because the epitope can be detected in denatured samples by Western blot and pulled down in unfixed co-IPs. This also suggests that other subunits in the complexes have different epitope availability and show different cross-linking efficiencies for the same subunit. Our previous study used a custom-made rabbit polyclonal antibody to BAF155 to show the recruitment of another subunit within the SS18-containing complexes (15). Unfortunately, the quantities of this antibody needed for this study are no longer available, nor have we previously tested its ability to detect pBAF recruitment. We attempted ChIPs using commercially available antibodies to BAF155 and BRG1, present in all mESC BAF complexes, without success. So, we could only show the enrichment of other subunits not found in all BAF complexes, such as BAF57, ARID1A, and SS18, making direct comparisons of the relative enrichment of the different BAF complexes difficult (Fig S5). However,

these data do show statistically significant enrichment of a non-tagged subunit for each of the three BAF complexes.

A second potential limitation of the study is the apparent high background level of H2AK119ub across the genome (23). Whereas we did not anticipate this affecting our study of looking at H2AK119ub at a specific locus before and after recruitment of a BAF complex, the H2AK119ub ChIPs were more variable than the H3K27me3 ChIPs. They may not be as good of a readout for PRC1-mediated facultative heterochromatin complexes as RING1B is itself, and more weight could be placed on RING1B. However, there may be an unknown biological reason for the discordance between RING1B and its apparently more persistent histone mark.

# Materials and Methods

### Cell culture

HEK293T cells were obtained from ATCC (CRL-3216) and cultured using standard parameters using DMEM media (11960044; Gibco) containing 10% FBS (Omega), 1% GlutaMAX (35050061; Gibco), and 1% Penicillin-Streptomycin (15140122; Gibco). mESCs were cultured using standard culture procedures. These cells were isolated by Jerry Crabtree's laboratory (Stanford University). Cells were maintained feeder-free using KnockOut DMEM (10829018; Gibco) media containing, 7.5% KnockOut Replacement Serum (10828028; Gibco), 7.5% ES-sure FBS (ASM-5017; Applied StemCell), 1% Penicillin-Streptomycin (15140122; Gibco), 1% GlutaMAX (35050061; Gibco), 1% MEM Non-Essential Amino Acids (11140050; Gibco), 1% Sodium Pyruvate (11360070; Gibco). LIF was replaced daily, and ES cells were passaged every 48 h.

### Lentivirus transduction

HEK 293T cells were transfected with lentiviral constructs and packaging plasmids (psPAX2 and pMD2.G) using PEI transfection (Polysciences). 2 d post-transfection, the virus-containing cell culture media was collected, filtered with a 0.44 $\mu$m syringe filter, and centrifuged at 50,000 $g$ for 2.5 h at 4°C (SW28 rotor on ultracentrifuge). The viral pellet was resuspended in PBS and used for subsequent infections. Selection of lentiviral constructs was achieved with the following doses: puromycin (2 $\mu$g/ml), blasticidin (10 $\mu$g/ml), hygromycin (150 $\mu$g/ml), and zeocin (200 $\mu$g/ml).

### Antibodies used in this study

Rabbit polyclonal antibody α-PHF10 (PA5-30678; Invitrogen); Co-IP, ChIP, WB

Rabbit α-ARID2 (PA5-35857; Invitrogen); Co-IP, WB

Mouse α-V5 Tag (E9H8O) (#80076; Cell Signaling Technology); Co-IP, WB

Rabbit α-V5 Tag (D3H8Q) (#1302; Cell Signaling Technology); ChIP, WB

Rabbit α-ARID1A (D2A8U) (#12354; Cell Signaling Technology); ChIP

Rabbit α-H3K27me3 (C36B11) (#9733; Cell Signaling Technology); ChIP; WB

Rabbit α-SS18 (D6I4Z) (#21792; Cell Signaling Technology); ChIP rabbit polyclonal α-BAF60A (#A301-595A; Bethyl Laboratories); ChIP

Rabbit polyclonal α-H3 c-terminal (#13-0001; Epicypher); ChIP

Mouse α-H3 (#3638S; Cell Signaling Technology); WB rabbit α-H2AK119ub (D27C4) (#8240; Cell Signaling Technology); ChIP; WB

Rabbit α-SMARCC1/BAF155 (D7F8S) (#11956S; Cell Signaling Technology); WB

Mouse α-BRG1 (H-10) (sc-374197; Santa Cruz Biotechnology); WB

Rabbit polyclonal α-BAF57 (A300-810A; Bethyl Laboratories); ChIP

Rabbit recombinant α-Histone H3.3 (91191; Active Motif); ChIP, CUT&TAG

Rabbit α-Histone H4 (D2X4V) (#14149H4; Cell Signaling Technology); ChIP

Rabbit polyclonal α-SUZ12 (No: 39057; Active Motif); ChIP

Rabbit α-RING1B (D22F2) XP (mAb #5694; Cell Signaling Technology); ChIP

Mouse HA-antibody (2367S; Cell Signaling Technology); WB

Rabbit α-Frb (gift from Gerald Crabtree); WB

Goat anti-mouse IgG IRDye 680RD polyclonal secondary antibody (Li-Cor)

Goat anti-mouse IgG IRDye 800CW polyclonal secondary antibody (Li-Cor)

Goat anti-rabbit IgG IRDye 680RD polyclonal secondary antibody (Li-Cor)

Goat anti-rabbit IgG IRDye 800CW polyclonal secondary antibody (Li-Cor)

### Ammonium sulfate protein extraction

Cells were plated onto 15 cm$^2$ plates, and 30 × 10$^6$ cells were harvested after 48 h. After a single wash with PBS to remove any remaining media, the cells were lysed in 10 ml of the lysis buffer (10 mM Hepes pH 7.5, 25 mM KCl, 1 mM EDTA, 0.1% NP-40, 10% glycerol, 1 mM DTT, protease inhibitor tablet [Roche], 1 mM sodium orthovanadate, and 10 mM sodium butyrate) and incubated for 10 min on ice. Washing with lysis buffer one more time, lysed cells were resuspended into 660 $\mu$l of the resuspension buffer (10 mM Hepes pH 7.5, 100 mM KCl, 1 mM EDTA, 3 mM MgCl$_2$, 10% glycerol, 1 mM DTT, protease inhibitors cocktail [10 $\mu$g/ml of Leupeptin, Chymostatin, Pepstatin A], 1 mM sodium orthovanadate, 10 mM sodium butyrate and 300 mM ammonium sulfate) and incubated for 30 min at 4°C. After moving into the centrifugation tube (Beckman Coulter polycarbonate tube #343778), it is spun down with ultracentrifugation (Thermo Fisher Scientific rotor S140-AT) at 536,480$g$ for 10 min. The supernatant was incubated with 200 mg solid ammonium sulfate for 45 min at 4°C and spun down again with ultracentrifugation at 536,480$g$ for 10 min. The pellets are frozen and stored at −80°C.

### Co-immunoprecipitation

Protein pellets were resuspended in 220 $\mu$l of IP buffer (20 mM Hepes pH 7.5, 150 mM KCl, 1 mM EDTA, 1 mM MgCl$_2$, 10% glycerol, 0.1% Triton X-100, 1 mM DTT, protease inhibitors cocktail, 0.2 mM sodium orthovanadate, 10 mM sodium butyrate). Protein G beads washed

with PBS are incubated with each antibody, mouse α-V5 Tag (E9H8O) (1:50; Cell Signaling Technology), rabbit α-PHF10 (1:100; Invitrogen), and rabbit α-ARID2 (1:100; Invitrogen) in PBS for 1 h at RT. After incubation, protein G beads are washed with PBS two times and washed with an IP buffer two times. Protein G beads were incubated with 400 $\mu$l of protein extracts with the protein concentration of 0.75 $\mu$g/$\mu$l overnight at 4°C. Supernatants were saved as a flow-through, and the beads were washed with an IP buffer five times on ice. IPs were extracted with 50 $\mu$l of RIPA buffer (50 mM Tris–pH 7.8, 150 mM NaCl, 1% NP-40, 1% SDS, 0.1% sodium deoxycholate (DOC), 1 mM DTT, and protease inhibitor cocktails) by boiling for 5 min.

## Chromatin immunoprecipitation

ChIP samples were prepared by fixing one of two ways and then processed the same after the fixation steps.

### Single formaldehyde fixation
Cells were harvested by first washing with 10 ml PBS (Thermo Fisher Scientific) and then incubated in trypsin-EDTA 0.25% (Thermo Fisher Scientific) for 5 min. They were then washed with PBS and fixed using a final concentration of 1% formaldehyde for 10 min. The addition of 0.125% glycine then quenched fixation. This method was used for all samples except the ones listed below.

### Double fixation method
Detailed methods (an adaptation of Bing and Brasier (32)) are available at (33). This fixation method was used to map the enrichment of other BAF subunits at the recruitment locus (Fig S2).

### Preparation of chromatin and IP
Cell pellets (30 × 10$^6$ cells) were resuspended in CiA NP-Rinse 1 (50 mM Hepes pH 8.0, 140 mM NaCl, 1 mM EDTA, 10% glycerol, 0.5% NP-40, 0.25% Triton X-100) and incubated on ice for 10 min before being spun down at 1,000 $g$ for 5 min at 4°C. The supernatant was aspirated, and cells were resuspended in CiA NP-Rinse 2 (10 mM Tris–pH 8.0, 1 mM EDTA, 0.5 mM EGTA, 200 mM NaCl) and spun down at 1,000 $g$ for 5 min at 4°C. The supernatant was aspirated, and the salt was washed from the sides of the tube by gently adding 5 ml of Covaris shearing buffer (0.1% SDS, 1 mM EDTA pH 8.0, 10 mM Tris–HCl pH 8.0) along the ridge of the tube when rotating clockwise so that the buffer trickled down the sides and did not disturb the pellet. The samples were spun down at 1,000 $g$ for 3 min at 4°C, and the supernatant was carefully aspirated. This step was repeated one additional time before the pellet was resuspended in 0.9 ml of CiA Covaris shearing buffer (0.1% SDS, 1 mM EDTA pH 8.0, 10 mM Tris–HCl pH 8.0) + protease inhibitor cocktail (1:1,000) and transferred to a Covaris glass tube.

### Sonication
Cells were sonicated for 8 min (single fixation ChIP) or 10 min (double fixation ChIP) to generate DNA fragments of the desired size using a Covaris E220 Evolution at 5.0 duty factor, 140 peak power, and 200 cycles per burst. After this, samples were transferred to microcentrifuge tubes and spun at 10,000 $g$ for 5 min at 4°C, and the supernatant (chromatin stock) was transferred to a new tube. Then, 25 $\mu$l of chromatin stock was aliquoted as input DNA. All sonicated chromatin was stored at −20°C until ready for IP.

### Immunoprecipitation
Sonicated chromatin was diluted in 5x IP buffer (250 mM Hepes, 1.5 M NaCl, 5 mM EDTA pH 8.0, 5% Triton X-100, 0.5% DOC, and 0.5% SDS) to a concentration of 1x IP buffer and incubated overnight with 10 $\mu$l Pierce protein A/G beads (Thermo Fisher Scientific) at 4°C with rotation. Beads were collected on a magnet and washed with 1x IP buffer for 3–5 min with rocking. Beads were then washed with 0.5 ml DOC buffer (10 mM Tris–pH 8, 0.25 M LiCl, 0.5% NP-40, 0.5% DOC, 1 mM EDTA) for 1–3 min with rocking and then washed with 1 M TE pH 8.0. The residual supernatant was removed.

### Reverse cross-linking
Beads incubated as part of IP were resuspended and mixed well in 100 $\mu$l TE, 5 $\mu$l 10% SDS, and 5 $\mu$l Proteinase K (10 mg/ml). Inputs were resuspended and mixed well in 75 $\mu$l TE, 5 $\mu$l 10% SDS, and 5 $\mu$l Proteinase K (10 mg/ml). Samples were then incubated at 55°C for 3 h, followed by an incubation at 65°C overnight.

### Elution
Supernatant was collected from each sample and transferred to a new tube. Then, 550 $\mu$l of NTB Binding Buffer (Machery Nagel) was added to each sample, and samples were loaded into NucleoSpin PCR/gel clean-up columns (Machery Nagel). Columns were washed with 750 $\mu$l of Buffer PE (80% EtOH, 10 mM Tris–pH 7.5). IPs were eluted with 25 $\mu$l of Buffer EB (Machery Nagel), and inputs were eluted with 50 $\mu$l of Buffer EB.

### ChIP-qPCR
pPCR samples were prepared using Accuris qMAX SYBR Green (MidSci), according to the manufacturer's instructions. Analysis of qPCR samples was performed on a QuantStudio 3 Flex system (Life Technologies). For ChIP-qPCR experiments, enrichment (bound over input) values were normalized to values with no RAP treatment (Rap/No Rap) and then to a control locus (HoxA9) enriched for polycomb-associated histone marks.

## Transcriptional analysis

Cells were plated onto 6 cm$^2$ plates and left to attach overnight. For FIRE-dCas9 lines, 3 nM rapamycin was added to experimental plates, and the vehicle was added to control plates 24 h before RNA isolation using TRIsure (Bioline) per the manufacturer's instructions. Before TRIsure treatment, cells were visualized under the microscope to ensure subconfluency and stem cell morphology. cDNA was made from 1 $\mu$g of total RNA using the Superscript VIVO mix (Invitrogen). cDNA was diluted 1:4 and 2 $\mu$l was used in a 20 $\mu$l qPCR TaqMan assay (Invitrogen). Assay probes were used to detect cDNA from Nkx2-9 (Mm00435145_m1). Ywhaz (Mm01722325_m1) was used for normalization (34). An mESC cell line constitutively expressing dCas9-VP64 (24) with the same sgRNAs to Nkx2-9 used in the FIRE-dCas9 experiments was used as a positive control.

## Western blots

Whole-cell extract of each sample was prepared using RIPA Buffer (1% SDS), protease inhibitor cocktail (1:1,000), 1 M DTT (1:1,000), and Benzonase nuclease (Sigma-Aldrich) (1:200). Proteins were then separated by SDS–PAGE electrophoresis with a 4–12% Bis-Tris

protein gel (Thermo Fisher Scientific) in 1x MOPS SDS Running Buffer (Thermo Fisher Scientific). Bands were transferred to an Immobilon-FL PVDF membrane (Thermo Fisher Scientific), then blocked in Intercept Protein-Free Blocking Buffer (Li-Cor) for 1 h. The membrane was then incubated overnight with Primary Antibodies diluted (1:1,000) in Intercept T20 Antibody Diluent (Li-Cor). The membrane was washed four times in TBS-T (0.2% Tween 20) for 5 min in each wash and probed with IRDye fluorescence-conjugated secondary antibody (Li-Cor) that was diluted (1:20,000) in T20 Antibody Diluent (Li-Cor) with 0.01% SDS. After, the membrane was washed four times in TBS-T (0.2% Tween 20) before a final rinse in TBS to remove the residual Tween 20. Bands were detected using an Odyssey DLx imaging system (Li-Cor). Raw Western blots used in Figs are provided as source files. As a loading control, we have used total protein staining. The staining method followed company protocols (Revert 700 Total Protein Stain Kits, Li-Cor): the membrane is stained with total protein staining reagent for 5 min, washed with wash buffer for 30 s twice, and scanned. Destaining of total protein stain is performed using a destaining buffer (Li-Cor) for 5 min. Then the blot is processed as usual with incubation in blocking buffer and antibody staining as described above.

### EZH2 inhibition

To inhibit the polycomb repressive complex 2 protein, we used a specific chemical inhibitor to EZH2 called GSK126 (Cat. No. 6790; Tocris). GSK126 was added to cell culture media at 3 $\mu M$ and was replaced every 24 h when either cell culture media was changed or cells were split. Cells were harvested for Western blots on day 0, day 2, and day 4 under both DMSO vehicle control and GSK126 conditions, and whole cell extract was made as described above. Cells were harvested on day 4 for RNA as described above. Finally, cells were harvested and fixed on day 4 for ChIP-qPCR as described above (single formaldehyde fixation).

### CUT&TAG

CUT&TAG direct was performed as previously described (35). Customizations to the published protocol include: 35,000 TC1 mESCs were used in each reaction. H3.3 antibody (Active Motif 91191) was used at 1:50 dilution and incubated overnight. Libraries were run on a TapeStation, and libraries were quantified using the NEBNext Library Quant Kit for Illumina (E7630L) and pooled in equal molar ratios. Samples were sequenced at the OMRF Clinical Genomics Core on a NovaSeqX. Quality control of samples was performed with fastqc (v0.12.1) (36) and fastq screen (v.0.15.3) (37). Reads were aligned using bowtie2 (v.2.5.0) (38) (--end-to-end --very-sensitive --no-mixed --no-discordant --phred33 -I 10 -X 700 -p 8). SAMtools (v1.18) (39) was used to convert SAM to BAM files and to generate raw and scaled bigwig files. Peaks were called using macs2 (v.2.2.7.1) (40) (-q 0.1 --keep-dup all --nomodel 2). Browser tracks were visualized in Integrative Genomics Viewer (41).

### Plasmids used or generated in this study

All plasmids used in this study were confirmed by nanopore-based whole plasmid sequencing (Plasmidsaurus or OMRF Clinical Genomics Center). The generated plasmids will be submitted to Addgene upon peer-reviewed publication of this manuscript but are immediately available by contacting the corresponding author. MS2–stem-loop sgRNAs targeting *Nkx2-9* were previously described (15).

### Primers used in this study

JK800 Nkx2-9 (−2,455) AAA TGA CCG GGC TCT GTA TG.
JK801 Nkx2-9 (−2,455) AGT TCC CGC TTC ACA TTC TC.
JK429 mNkx2-9 (−820) CTC CAT TCG AGG ACC CAA GG.
JK430 mNkx2-9 (−820) CTG CTA ACT GGC ACC GAC TT.
JK431 mNkx2-9 (−315) TCT TGG GTG GCG AAC AGT G.
JK432 mNkx2-9 (−315) AAT AAA GTC GCT CCA CCC TCC.
JK433 mNkx2-9 (+160) CCG CTC CTA AGG ATG GAA GT.
JK434 mNkx2-9 (+160) TTC AAA GCC CTC CGA GTA GC.
JK435 mNkx2-9 (+630) ATC CCG GTC TTT TCG GAT CG.
JK436 mNkx2-9 (+630) TGC GTC TGA GTC CAC ACA TC.
JK437 mNkx2-9 (+1,400) ACC TCT GCC GTT GTT GCT C.
JK438 mNkx2-9 (+1,400) GCC TTC GGA TAT GGC AGC AT.
JK342 mGapdh F ChIP CTC TGC TCC TCC CTG TTC C.
JK343 mGapdh R ChIP TCC CTA GAC CCG TAC AGT GC.
JK344 mHoxa9 F ChIP AAG AAG GAA AAG GGG AAT GG.
JK345 mHoxa9 R ChIP TCA CCT CGC CTA GTT TCT GG.

# Supplementary Information

# Acknowledgements

This work was supported in part by grants from NIH/NIGMS 5P20 GM103636 (Research Project Leader: JG Kirkland), the Stephen M. Prescott Endowment Fund for the Best and Brightest (JG Kirkland), and an OMRF pre-doctoral scholarship (J Park). Data processing and analysis were supported by the OMRF Center for Biomedical Data Sciences and Kevin Boyd in the Cell Cycle and Cancer Biology Program.

### Author Contributions

M Bergwell: data curation, formal analysis, investigation, visualization, methodology, and writing—review and editing.
J Park: data curation, formal analysis, investigation, methodology, and writing—review and editing.
JG Kirkland: conceptualization, data curation, formal analysis, supervision, funding acquisition, investigation, visualization, methodology, and writing—original draft.

### Conflict of Interest Statement

The authors declare that they have no conflict of interest.

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
