## [Reviewer comments · Life Science Alliance]

Life Science Alliance

Differential Modulation of Polycomb-Associated Histone Marks by cBAF, pBAF, and gBAF Complexes

Mary Bergwell, JinYoung Park, and Jacob Kirkland

DOI: 10.26508/lsa.202402715

Corresponding author(s): Jacob Kirkland, Oklahoma Medical Research Foundation

Review Timeline:

Submission Date:	2024-03-13
Editorial Decision:	2024-05-01
Revision Received:	2024-08-14
Editorial Decision:	2024-08-15
Revision Received:	2024-08-20
Accepted:	2024-08-21

Transaction Report:

May 1, 2024

Re: Life Science Alliance manuscript #LSA-2024-02715-T

Dr. Jacob G Kirkland
Oklahoma Medical Research Foundation
Cell Cycle and Cancer Biology
825 N.E. 13th Street
MS-48
Oklahoma City, OK 73104

Dear Dr. Kirkland,

Thank you for submitting your manuscript entitled "Differential Modulation of Polycomb-Associated Histone Marks by cBAF, pBAF, and gBAF Complexes" to Life Science Alliance. The manuscript was assessed by expert reviewers, whose comments are appended to this letter. We invite you to submit a revised manuscript addressing the Reviewer comments.

Thank you for this interesting contribution to Life Science Alliance. We are looking forward to receiving your revised manuscript.

Sincerely,

B. MANUSCRIPT ORGANIZATION AND FORMATTING:

Reviewer #1 (Comments to the Authors (Required)):

Bergwell and colleagues previously reported that artificial recruitment of SWI/SNF at polycomb target leads to a rapid loss of polycomb-associated histone modifications. Here, they pursued this study to determine whether the three primary forms of mSWI/SNF are all required for this process or not. To tackle this question, they artificially tethered different subunits that are each specific of a complex (DOF2, BRD9 and PHF10). They subsequently studied the consequences of this recruitment on chromatin.

However, no robust conclusion can be made as long as the authors have not shown that the three baits recruit SWI/SNF to the same extent. The authors mentioned some difficulty in getting an antibody recognizing subunit common to all SWI/SNF complex, however this is mandatory for this study.

Specific comments:

Figure 1:

- "We finally transduced these lines with three sgRNA with extra stem-loop structures targeting the promoter of Nkx2.9 to complete the system." This sentence needs to be reworded to avoid giving the impression that the stem-loop structures provide sequence specificity for recruitment to the Nkx2.9 promoter, when in fact only the sgRNA spacer sequences do so.
- Co-IPs in panels E and F are not controlled. Need a condition where the bait is absent in the input to show that the IP doesn't recover the proteins nonspecifically. Loading controls should be included in Panels A and D.
- Ideally the authors should show not only incorporation of the tagged subunits into the respective BAF complexes, but also genetic complementation assays. Do the tagged proteins rescue the cellular phenotypes observed upon deletion of the corresponding endogenous factors?

Figures 2, 3, 4:

- It is very difficult to rule out a trivial explanation for the authors' results: perhaps gBAF less effectively opposes Polycomb complex activity than cBAF because it is less robustly recruited to the Nkx2.9 gene promoter. The authors should try to modulate the strength of cBAF recruitment to mimic the gBAF situation, in order to check that the effects they observe are in fact dependent on the identity of the BAF complex rather than the degree to which it binds their target locus.
- It is very difficult to evaluate ChIP result presented as a ratio normalized over another localization. Percent of input should be provided (with the no RAP control on the side). This is particularly important considering the high variability in Figure 1A for instance.

Figure 5:

- "Curiously, we also saw an increase in H3.3 even though we didn't see a large loss of polycomb-associated histone marks after the recruitment of gBAF (Figure 3B and E)." In fact this is not so surprising, because if the initial levels of H3.3 are extremely low, than even a modest change in absolute levels, such as might be compatible with the results presented in Figures 3 and 4, could translate to a substantial fold increase.

Figure 6:

- The authors appear to conclude that artificial recruitment of cBAF to the Nkx2.9 promoter leads to greater transcriptional activation than gBAF or pBAF because of the ability of cBAF to trigger the removal of Polycomb-associated histone marks. But no evidence of causality is presented. In order to demonstrate the link, the authors should perform the assay in conditions in which PRC2 is disrupted. According to their model they should obtain activation at least equivalent to that provided by cBAF recruitment, and in this scenario recruitment of any of the three BAF complexes should not lead to any additional activation.

Reviewer #2 (Comments to the Authors (Required)):

"Differential Modulation of Polycomb-Associated Histone Marks by cBAF, pBAF, and gBAF Complexes" by Mary Bergwell et al is an excellent paper utilizing the authors' unique systems to reveal the eviction of polycomb-associated histone is specifically

executed by canonical BAF. This finding is very interesting and could make a substantial contribution to understand the precise mechanisms of competition between epigenetic activators and repressors. However, some of the data could be improved to support authors' conclusion more efficiently.

Major

1. In Figure 3, discrepancy between effect on H2AK119Ub and H3K27me3 is very interesting, however in a sense different from prior research. For example, Stanton BZ et al (ref. 9) revealed that deletion of SMARCA4 induced upregulation of PRC1 occupancy and SMARCA4 without ATPase activity failed to evict PRC1, indicating cBAF has a function to compete with PRC1. Therefore, to strengthen the authors' claim, more convincing data would be required. The authors should show the level of H3K27me3 and H2AK119Ub at negative loci. Also, as I aware that ChIP for H2AK119Ub often encounters higher background, adding ChIP for RING1B could make the situation more persuasive.
2. In Figure 4, I'm not sure if ChIP for H3 could be applicable to accurately estimate nucleosome positioning and density. To discuss nucleosome occupancy, MNase-seq could be more convincing.

Minor

1. In Figure 1 to 5, the result of ChIP qPCR should include the ratio to input samples.
2. Related to Figure 3, the authors should perform ChIP for PRC1 (RING1B) and PRC2 (SUZ12 or EED).
3. Related Figure 3, the difference between the current study and ref. 9 should be discussed.
4. In Discussion, (1) Co-recruitment of an enzyme or enzymes that remove the marks, leaving the rest of the nucleosomes intact, an article Takano, J. et al. PCGF1-PRC1 links chromatin repression with DNA replication during hematopoietic cell lineage commitment. *Nat Commun* 13, 7159 (2022). <https://doi.org/10.1038/s41467-022-34856-8> could be supportive as in this paper the authors revealed SWI/SNF complexes compete with PRC1/2 to evict H3K27me3 without involving the recruitment of demethylase in hematopoietic progenitor cells.

We want to thank both reviewers for their timely reviews and the time and thought they dedicated to our manuscript. We feel strongly that suggestions from both reviewers, in the additional experiments proposed and in the discussion of our results, made this a much better paper. A point-by-point response to how we have responded to comments is below in blue text.

Reviewer #1 (Comments to the Authors (Required)):

Bergwell and colleagues previously reported that artificial recruitment of SWI/SNF at polycomb target leads to a rapid loss of polycomb-associated histone modifications. Here, they pursued this study to determine whether the three primary forms of mSWI/SNF are all required for this process or not. To tackle this question, they artificially tethered different subunits that are each specific to a complex (DOF2, BRD9 and PHF10). They subsequently studied the consequences of this recruitment on chromatin.

However, no robust conclusion can be made as long as the authors have not shown that the three baits recruit SWI/SNF to the same extent. The authors mentioned some difficulty in getting an antibody recognizing subunit common to all SWI/SNF complexes, however this is mandatory for this study.

Specific comments:

Figure 1:

- "We finally transduced these lines with three sgRNA with extra stem-loop structures targeting the promoter of *Nkx2.9* to complete the system." This sentence needs to be reworded to avoid giving the impression that the stem-loop structures provide sequence specificity for recruitment to the *Nkx2.9* promoter, when in fact only the sgRNA spacer sequences do so.

We thank the referee for pointing out this confusing sentence. We did not mean to imply that the stem-loops provide any sequence specificity and have replaced the sentence with the following:

"We finally transduced these lines with three sgRNA targeting the promoter of *Nkx2.9* that also contain extra stem-loop structures to complete the system."

- Co-IPs in panels E and F are not controlled. Need a condition where the bait is absent in the input to show that the IP doesn't recover the proteins nonspecifically. Loading controls should be included in Panels A and D.

We have performed IP controls on wild-type mESCS, showing that a pulldown with V5 in these cells that don't express a V5 epitope does not pull down BAF subunits, showing the specificity of our co-IP in Figure E. This control is now presented as new Figure 1F.

For Figure F (now G), we attempted to build a degradable allele of ARID2 since we do not have a mouse cell line lacking ARID2. This process took two months, and we could only get one homozygous clone out of many heterozygous clones. However, when characterizing this clone, we realized it did not degrade ARID2-dTAG-V5 as expected, even though the genetic tag was made in-frame. We realized that the tagged version of ARID2 was running slightly smaller than the WT ARID2 in unedited cells instead of slightly larger. This led us to hypothesize that the

ARID2-dTAG fusion is deleterious to the cell (even without adding the degradation molecule) and the clone adapted to express a slightly truncated version of the ARID2 protein that does not contain the dTAG-V5 insertion and some unknown amount of the C-terminus of the endogenous ARID2 protein. Therefore, we could not devise a cell line where the antibody epitope to ARID2 was absent. We are attempting to try an N-terminal tag for ARID2, but this will take an additional 2 months, and we are unsure if it will be successful.

Additionally, siRNA knockdowns of ARID2 in mESCs are insufficient for this control, as enough left-over protein will be present to be enriched in a pull-down assay. We are unaware of any ARID2 knock-out cell lines in other labs. While having this control is ideal, we failed in our attempt to build a suitable system to perform it.

We repeated the western blots in Figures 1C and 1D using new samples to include total protein loading controls. Therefore, both of these figures have been replaced with new images. The new raw uncropped images are included in Figure S1.

Figure for reviewers: western blot of WT TC1 cells and a homozygous CRISPR knock-in of a c-terminal degron tag (dTAG)-V5 fusion into the endogenous *Arid2* locus in mESCs (clone 3). This clone was treated with DMSO vehicle or the degron molecule d13 for 24 hrs, however no degradation occurred.

- Ideally the authors should show not only incorporation of the tagged subunits into the respective BAF complexes, but also genetic complementation assays. Do the tagged proteins rescue the cellular phenotypes observed upon deletion of the corresponding endogenous factors?

While we agree that a phenotypic rescue of PHF10, DPF2, and BRD9 would be a gold-standard control, we do not know what phenotype to look for to rescue. In fact, we proposed to look at what happens to mESCs in the absence of PHF10, DPF2, and BRD9 in a grant application, but we have not completed these experiments and, therefore, can't do genetic complementation assays in mESCs.

Figures 2, 3, 4:

- It is very difficult to rule out a trivial explanation for the authors' results: perhaps gBAF less effectively opposes Polycomb complex activity than cBAF because it is less robustly recruited to the *Nkx2.9* gene promoter. The authors should try to modulate the strength of cBAF recruitment to mimic the gBAF situation, in order to check that the effects they observe are in fact dependent on the identity of the BAF complex rather than the degree to which it binds their target locus.

We do not disagree with the reviewer's point and tried to discuss it fully in our "Limitations" section of the manuscript. To address this better, we looked at the correlation of V5 enrichment when recruiting cBAF complexes to the *Nkx2-9* locus versus H3K27me3 levels. If the eviction of H3K27me3 were strongest in the samples where the V5 recruitment was highest, we would expect to see a strong anti-correlation between these two proteins. In these graphs, we took the

V5 value for each sample at the -315 locus (peak recruitment site) and plotted it against the H3K27me3 for each matched sample at each of the eight loci. However, we did not find the expected anti-correlation. We either found no correlation or a slight positive correlation (the opposite of expected if this theory is correct) between the level of V5 (cBAF) recruitment and the loss of H3K27me3. Additionally, since we report the data as bound/input enrichment rather than a fold-change, as the referees suggested, the difference between DPF2-V5 and BRD9-V5 is less and contains overlapping

Supplemental Figure 3: dot plots of bound/input (b/i) values after rapamycin-based recruitment of cBAF complexes via a DPF2-2xFkbp-V5 protein fusion. The V5 b/i value at -315 (recruitment site) is plotted against the b/i of H3K27me3 at each of the 8 primer pairs tested for all five replicates. A linear regression correlation line and R² value is provided for each of the plots.

data points (See Figure 3). This data is now presented in Supp Figure 2. Similar results were found using the b/i values at the +160 locus compared to H3K27me3 at all eight loci.

While we cannot eliminate the idea that a BAF complex must exceed some minimum threshold to evict polycomb-associated histone marks efficiently, our data suggests that once that minimum threshold is met, eviction efficiency is not directly correlated with the amount of BAF at the locus. It is possible that the different complexes have different minimum thresholds, but we have not identified a good way to test this hypothesis. However, if we compare the bound/input enrichments of cBAF and gBAF, we see overlapping data points. While the median for cBAF is

higher than gBAF, this is not a statistically significant difference (Figure S2A)

Supplemental Figure 2: Chromatin immunoprecipitations against various antibodies to determine enrichment for other BAF subunits during recruitment at the *Nkx2.9* locus. (A) enrichment of V5 with RAP conditions cBAF versus gBAF shows no statistical difference in recruitment enrichment. (B) V5 enrichment with pBAF recruitment via PHF10-2xFkbp-V5 under control and Rapamycin-based recruitment (RAP) conditions (C) V5 versus PHF10 enrichment with pBAF recruitment via PHF10-2xFkbp-V5 after Rapamycin addition. Data is presented as bound/input either no recruitment or recruitment (n=4-5; s.e.m.)

Finally, we are no longer convinced that even using the same antibody is a reliable way to compare enrichments across different proteins and complexes. For the pBAF complex, we recruit using a PHF10-2xFkbp-V5 protein fusion. However, when using the same V5 antibody, we see no enrichment, but we do see robust enrichment with a PHF10 antibody. This suggests that protein and complex specific conformations, even for orthologous proteins, can have different effects on epitope availability and/or cross-linking effects. See Supplemental Figures 2B and 2C.

We added/edited sections of the manuscript to now read:

Results section:

cBAF has a higher median enrichment than gBAF, but these two datasets have overlapping data points and are not statistically significant from each other (Supplemental Figure 2A). However, PHF10-based dd pBAF recruitment failed to show enrichment via the V5 antibody (Supplemental Figure 2B). We then performed ChIP against the PHF10 subunit itself and found robust enrichment of the recruited subunit (Figure 2C, Supplemental Figure 2C). This observation complicates our ability to directly compare the overall recruitment levels of the different complexes since even the same antibody shows a differential ability to bind the epitope (V5 in this case) depending on the subunit tagged and the complex it incorporates into. However, in each case, enrichment of the recruited subunit is specifically around the sgRNA binding sites (spanning -172 to -25 bp) and does not spread far up or downstream.

Limitations section:

One limitation of this study is that it proved impossible to measure the recruitment rates of the three BAF complexes in relation to each other. While all complexes contained a V5-tagged subunit, using a V5 antibody in the PHF10-2xFkbp-V5-based pBAF recruitment experiments showed no enrichment, whereas an antibody against PHF10 did. This suggests the V5 antibody had variable accessibility to its epitope within a formaldehyde-fixed BAF complex. We hypothesize that the V5 epitope is masked within a pBAF complex by other subunits after fixation since the epitope can be detected in denatured samples by western blot and pulled down in unfixed co-IPs. This also

suggests that other subunits in the complexes have different epitope availability and show different cross-linking efficiencies for the same subunit.

...

So, we could only show the enrichment of other subunits not found in all BAF complexes, such as BAF57, ARID1A, and SS18, making direct comparisons of the relative enrichment of the different BAF complexes difficult (Supplemental Figure 4). However, this data does show statistically significant enrichment of a non-tagged subunit for each of the three BAF complexes.

- It is very difficult to evaluate ChIP result presented as a ratio normalized over another localization. Percent of input should be provided (with the no RAP control on the side). This is particularly important considering the high variability in Figure 1A for instance. We have changed all figures to be presented as a ratio of the bound signal/input signal with separate data for untreated and RAP-treated samples side-by-side, as requested. While we originally felt that normalization over another unchanged positive locus was a valuable normalization step to control for variations in IP efficiencies similar to using a control gene (e.g. GAPDH) in an RT-qPCR experiment to control for the efficiency of cDNA conversions, we have removed that step at the reviewer's request. However, it was a mistake to include normalization in Figure 1A since there is no V5 positive control locus to properly normalize against. This actually contributed to an increase in variability, and removing this step decreased the variability, as one can see now in Figure 2.

Figure 5:

- "Curiously, we also saw an increase in H3.3 even though we didn't see a large loss of polycomb-associated histone marks after the recruitment of gBAF (Figure 3B and E)." In fact this is not so surprising, because if the initial levels of H3.3 are extremely low, then even a modest change in absolute levels, such as might be compatible with the results presented in Figures 3 and 4, could translate to a substantial fold increase. First we have changed the data presentation to show bound/input rather than a fold-change. Secondly, we performed CUT&TAG against H3.3 in two biological replicates in non-recruited conditions to set a baseline for the *Nkx2-9* locus compared to other regions of the genome. We found three peaks that encompass the *Nkx2-9* region we are interrogating in both replicates suggesting that H3.3 levels are not extremely low in this region. In fact, not only are there called peaks in this region, but the peaks are between the 18th and 48th percentile of all peaks in the genome. We have included browser tracks from the *Nkx2-9* locus in Supplemental Figure 5G. Upon better statistical testing on the b/i values, the gain in H3.3 from gBAF recruitment was not statistically significant, so we have removed this subjective phrasing from the manuscript. The levels of H3.3 are comparable to *Gapdh* and increase above *Gapdh* levels upon recruitment of cBAF complexes.

The results section on H3.3 now reads:

Since we found that nucleosome depletion was insufficient to explain the loss of polycomb-associated histone marks, we set out to test model (3), whereby the removal

of a polycomb-associated nucleosome is then replaced with an unmodified nucleosome. Outside the centromere, mammalian nucleosomes may contain one of three Histone 3 variants: H3.1/H3.2 (replication-dependent deposition) or H3.3 (replication-independent deposition). Here, we hypothesized that the replacement nucleosomes may be deposited independently of replication. H3.3 differs from H3.1 and H3.2 by 4 or 5 amino acids, so we used an antibody raised against this variable region and is specific to the H3.3 variant. Here, we found an increase in H3.3 histones after the recruitment of cBAF (Figure 5A). There was no significant gain in H3.3 after the recruitment of gBAF or pBAF complexes (Figure 5B and 5C). These data contrast the data when we used an antibody that recognizes all of H3.1, H3.2, and H3.3, whereby levels of H3 were slightly depleted or remained unchanged (Figure 4). The possibility remained that H3.3 levels at *Nkx2-9* were so small that a slight increase could lead to a large fold-change upon recruitment of BAF complexes. To address this, we performed CUT&TAG genome-wide on H3.3 and mapped peaks in two biological replicates (Supplementary Figure 5G). Here, we found that there were indeed three reproducible H3.3 peaks within the *Nkx2-9* locus, ranging from 18 to 48 percentiles of all peaks in the genome. Additionally, the H3.3 qPCR enrichment levels were similar to those at the *Gapdh* and *HoxA9* loci (Figure 5), suggesting *Nkx2-9* is not devoid of all H3.3 before recruiting BAF complexes. Therefore, these data are consistent with a model whereby cBAF evicts nucleosomes with polycomb-associated modifications, which are then replaced with H3.3 containing nucleosomes that largely lack H3K27me3 modifications.

New Figure 5:

Figure 5: cBAF recruitment leads to a gain in this histone variant H3.3. Chromatin immunoprecipitations followed by qPCR detecting enrichment of histones and their modifications. The recruitment site is approximately -100 bp. 0 bp denotes the TSS of *Nkx2-9*. Pulldown of H3.3 after recruitment of cBAF complexes. **(B)** Pulldown of H3.3 after recruitment of gBAF complexes. **(C)** Pulldown of H3.3 after recruitment of pBAF complexes. Data is presented as bound/input either no recruitment or recruitment (n=4; s.e.m.). Significance was evaluated by Multiple Mann-Whitney tests. * denotes $q < 0.05$

New Figure S5G:

G.

Supplemental Figure 5: Chromatin Immunoprecipitations (A) The ratio of H3K27me3 over H3 after recruitment of cBAF complexes (B) The ratio of H3K27me3 over H3 after recruitment of gBAF complexes (C) The ratio of H3K27me3 over H3 after recruitment of pBAF complexes (D) The ratio of H2AK119ub over H3 after recruitment of cBAF complexes (E) The ratio of H2AK119ub over H3 after recruitment of gBAF complexes (F) The ratio of H2AK119ub over H3 after recruitment of pBAF complexes (n=4-5; s.e.m.). Data is presented as bound/input either no recruitment (control) or recruitment (RAP) (n=5; s.e.m.). Significance was evaluated by Multiple Mann-Whitney tests. (G) Browser track of the *Nkx2-9* locus with H3.3 CUT&TAG based bigwig and called peaks tracks for n=2 biological replicates plotted.

Figure 6:

- The authors appear to conclude that artificial recruitment of cBAF to the *Nkx2-9* promoter leads to greater transcriptional activation than gBAF or pBAF because of the ability of cBAF to trigger the removal of Polycomb-associated histone marks. But no evidence of causality is presented. In order to demonstrate the link, the authors should perform the assay in conditions in which PRC2 is disrupted. According to their model they should obtain activation at least equivalent to that provided by cBAF recruitment, and in this scenario, recruitment of any of the three BAF complexes should not lead to any additional activation.

We thank the reviewer for pointing out that our model needed more clarity. We did not intend to imply that the removal of PRC2 is sufficient for transcriptional activation. However, because the reviewer understood this to be our model, we decided to test it. Thank you for suggesting this experiment. It added greatly to our study and allowed us to better describe our model. We used an EZH2 inhibitor (GSK126) for 2 and 4 days and assessed global H3K27me3/H2AK119Ub levels by western blot, H3K27me3/H2AK119Ub levels at *Nkx2-9* by ChIP-qPCR and transcription of the *Nkx2-9* gene by RT-qPCR. Here we found that H3K27me3 levels were below detection at both 2 and 4 days of EZH2 inhibition. We did not find any increase in *NKX2-9* mRNA by RT-qPCR after 4 days of EZH2 inhibition suggesting that loss of H3K27me3 and PRC proteins from the locus are not sufficient for transcriptional activation. However, it was possible that H3K27me3 levels remained normal at the *Nkx2-9* gene despite a global reduction in the mark, so we performed ChIP-qPCR, where we found that H3K27me3 was indeed greatly reduced at *Nkx2-9*. Inhibition of PRC2 complexes did not affect global H2AK119ub levels, nor did it affect H2AK119ub levels at *Nkx2-9*, suggesting that a loss of both marks or classes of PRCs may be required to activate the gene. However, these data are consistent with our

findings and others, which show that the BAF-mediated removal of PRC-associated histone marks or the PRC1 protein RING1B at *Ascl1* is not sufficient for transcriptional activation of gene¹⁻³. Together, these findings suggest that while the removal of PRC2 and its associated histone mark H3K27me3 is a necessary step for transcription, it is not sufficient. cBAF complexes must also remove PRC1 proteins and/or recruit unknown transcriptional activators or endogenously expressed transcription factors to aid in activating the gene.

We added this new section to the Discussion:

By inhibiting EZH2, the PRC2 methyltransferase responsible for writing the H3K27me3 mark in mESCs, we showed that simply removing H3K27me3 from most of the genome and at *Nkx2-9* gene specifically was not sufficient for transcriptional activation of *Nkx2-9*. This suggests that cBAF complex-mediated transcriptional activation is doing something other than just removing H3K27me3 from a locus to activate it. Because H2AK119ub remains at the locus and throughout the genome, it is possible that cBAF must remove both PRC1 and PRC2 proteins to activate a gene. These data are consistent with our findings, and those of others that the BAF-mediated removal of PRC-associated histone marks at *ASCL1* is insufficient for transcriptional activation of the gene^{8,9,15}. Therefore, the model we favor is that cBAF complexes must remove both PRC1/2 and bring in co-factors such as P300 or transcription factors to activate a gene. These co-factors are not accessing the promoter by removing H3K27me3 alone. It remains possible that the removal of PRC1/2 and H3K27me3, but not H2AK119ub, provides modulation of transcript levels, and removing all four classes (proteins and histone marks) could lead to even more robust transcriptional activation.

Reviewer #2 (Comments to the Authors (Required)):

"Differential Modulation of Polycomb-Associated Histone Marks by cBAF, pBAF, and gBAF Complexes" by Mary Bergwell et al is an excellent paper utilizing the authors' unique systems to reveal the eviction of polycomb-associated histone is specifically executed by canonical BAF. This finding is very interesting and could make a substantial contribution to understand the precise mechanisms of competition between epigenetic activators and repressors. However, some of the data could be improved to support authors' conclusions more efficiently.

Major

1. In Figure 3, discrepancy between effect on H2AK119Ub and H3K27me3 is very interesting, however in a sense different from prior research. For example, Stanton BZ et al (ref. 9) revealed that deletion of SMARCA4 induced upregulation of PRC1 occupancy and SMARCA4 without ATPase activity failed to evict PRC1, indicating cBAF has a function to compete with PRC1. Therefore, to strengthen the authors' claim, more convincing data would be required. The authors should show the level of H3K27me3 and H2AK119Ub at negative loci. Also, as I aware that ChIP for H2AK119Ub often encounters higher background, adding ChIP for RING1B could make the situation more persuasive.

We agree that this is somewhat different from previously reported data by Stanton et al. however, in the initial version of the manuscript, a direct comparison between the two studies

could not be made since we examined PRC1 through its histone mark H2AK119ub and Stanton et al. looked at the PRC1 protein RING1B. Now that we have done RING1B ChIPs when recruiting cBAF complexes, the two studies are in much better agreement. See further discussion below. However, we also believe that any differences that remain can be due to several factors. Changes could be locus-specific and/or dependent on the particular SMARCA4 mutation. SMARCA4 mutations that lead to a loss of ATPase activity also change the occupancy time of BAF complexes on chromatin, including some mutations that led to higher residency times on chromatin. Therefore, an ATPase-deficient complex may not be the same as a locus lacking a BAF complex at all.

We now show data at the *Gapdh* locus, which is actively transcribed with minimal polycomb-associated histone marks and polycomb proteins themselves, in addition to the *HOXA9* locus, which has high levels of PRC proteins and their histone marks. These loci show an expected dynamic range of PRC1 and H2AK119ub with the *Nkx2-9* locus lying in between.

We have also added ChIPs after the recruitment of cBAF for the PRC2 protein SUZ12 and the PRC1 protein RING1B, as requested by the reviewer. Here we see a significant loss of both SUZ12 and RING1B (Figure 3G-H) after the recruitment of cBAF complexes to *Nkx2-9*. This suggests that there is less discrepancy in the PRC proteins themselves than there is in the histone marks. This may be due to issues with a higher background with H2AK119Ub, as the referee suggests, or it could be a real biological phenomenon. However, the issue with a high background of H2AK119Ub was observed in genome-wide analysis, and it is suggested that the background is not artificial noise but is a real signal where H2AK119ub1 is found broadly across the genome⁴. Because we are only altering one locus and not the whole genome as done in the other studies that report this high background phenomenon, we feel we are able to directly compare H2AK119ub levels at this locus before and after cBAF recruitment because global levels of H2AK119ub1 remain unaltered. We added new sections to our Discussion and Limitations section regarding this issue.

Loss of a histone mark can occur through several mechanisms: (1) Co-recruitment of an enzyme or enzymes that remove the marks, leaving the rest of the nucleosomes intact; (2) Removal of a nucleosome, leaving a nucleosome-depleted region (NDR); or (3) Removal of a nucleosome which is then replaced with an unmodified nucleosome. Our data show that the cBAF-mediated loss of these marks, particularly H3K27me3, cannot be explained by nucleosome depletion alone (model 2), as more modified histones are removed following cBAF recruitment than total histones. While future work is required to distinguish between models 1 and 3, we currently have no data consistent with model 1. Others have shown that BAF competes with PRC complexes to evict H3K27me3 without the co-recruitment of demethylases in a hematopoietic stem cell system, consistent with model 3²⁵. These data suggest that the mode of H3K27me3 opposition may be shared across loci and cell types. For model 1 to be correct, cBAF must recruit KDM6A or KDM6B²⁶ to remove H3K27me3 marks and BAP1/ASXL1^{27,28} to remove H2AK119ub marks at the same sites. This co-recruitment seems unlikely compared to model 3, where cBAF removes polycomb-associated

nucleosomes, thereby removing H3K27me3 and H2AK119ub simultaneously. Following the removal of these nucleosomes, new nucleosomes containing H3.3 but lacking H3K27me3 are inserted into the chromatin, which cBAF complexes may have a lower affinity for removing once again compared to polycomb-associated H3.1 containing nucleosomes. This specificity may be through interactions with PRC1 proteins themselves⁹, which are bound at these modified nucleosomes and are not found at the unmodified nucleosomes, thereby instructing cBAF which nucleosomes are to be preferentially evicted. However, our observation that H3K27me3 is removed more efficiently and over a broader region than H2AK119ub suggests additional undiscovered mechanisms of PRC2 eviction. cBAF may also preferentially avoid evicting H3.3-containing nucleosomes regardless of the polycomb-associated histone modifications. The BAF subunits BRG1 and SMARCB1 interact with the H3.3 chaperone HIRA complex²⁹. This interaction may direct or promote H3.3 deposition at BAF-regulated sites, promoting transcription and a more open chromatin state³⁰. However, the BRG1 and SMARCB1 subunits are shared between all BAF complexes, so it remains unclear how different BAF complexes influence H3.3 deposition to varying degrees.

Additionally, we found a discrepancy between the PRC1 protein (RING1B) and the PRC1-associated mark H2AK119ub removal, where the actual PRC1 proteins are evicted much more efficiently. This could be a technical artifact behind H2AK119ub ChIPs or could be biological, where cBAF complexes physically interact with cPRC1 to evict the protein complex^{9,31}, but the histone mark remains more stable even without PRC1 present at the locus. An increase in PRC1 binding chromatin in BRG1 ATPase conditional knockout or mutant cells is consistent with our data showing cBAF complexes efficiently evict RING1B proteins, but the loss of H2AK119ub marks in BRG1 mutants was not explored in this study. Hence, we are unsure whether the difference between RING1B eviction and H2AK119ub is universal.⁹ This contrasts with PRC2 and its histone mark H3K27me3, which are both evicted with high efficiency and may speak to a difference in PRC1 versus PRC2 biology.

Limitations section addition: A second potential limitation of the study is the apparent high background level of H2AK119ub across the genome²³. While we didn't anticipate this affecting our study of looking at H2AK119ub at a specific locus before and after recruitment of a BAF complex, the H2AK119ub ChIPs were more variable than the H3K27me3 ChIPs. They may not be as good of a readout for PRC1-mediated facultative heterochromatin complexes as RING1B is itself, and more weight could be placed on RING1B. However, there may be an unknown biological reason for the discordance between RING1B and its apparently more persistent histone mark.

2. In Figure 4, I'm not sure if ChIP for H3 could be applicable to accurately estimate nucleosome positioning and density. To discuss nucleosome occupancy, MNase-seq could be more convincing.

While we agree with the reviewer that ChIP for H3 is inadequate for nucleosome positioning, we believe it is sufficient to describe the average nucleosome density over a region, just as we and

others have with modified histones^{1,2,5,6}. We tried to be careful with our wording and did not mean to imply a change in a particular positioned nucleosome (the +1 nucleosome, for example). We are measuring nucleosome occupancy over a 200-400bp region, and therefore, we are measuring a region that contains more than one nucleosome. While still controversial due to conflicting findings in yeast and in vitro, with mammalian studies, it is now proposed that mSWI/SNF (BAF) has a major role in opposing polycomb proteins and their associated histone marks rather than precise positioning of nucleosomes around promoters and transcriptional start sites. Miller et al. showed a lack of loss of global nucleosome positioning after the loss of SMARCA4 (Brg1 with losses only occurring at specific BAF binding sites likely in cooperation with TFs like STAT3⁵. Because *Nkx2-9* was not a BAF-specific nucleosome positioning site in mESCs, we took a broader look at average nucleosome occupancy/enrichment at various locations in the locus and only used a direct comparison of before and after recruitment of the various BAF complexes in these same regions contained within our qPCR amplicons.

Minor

1. In Figure 1 to 5, the result of ChIP qPCR should include the ratio to input samples.

As the reviewers requested, we are now presenting all ChIP data as bound/input ratios instead of a fold change. We agree that this is a clearer presentation despite our use of fold change in the past. Again we thank both reviewers for helping us communicate our findings in a clearer way.

2. Related to Figure 3, the authors should perform ChIP for PRC1 (RING1B) and PRC2 (SUZ12 or EED).

We have added additional ChIPs for RING1B and SUZ12 proteins when recruiting the cBAF complex into Figures 3G and 3H. This was a fairly major undertaking, but because of the reviewers' suggestions, our conclusions are now much more compelling. We thank the reviewer for this suggestion and for helping us strengthen our study. Details of this experiment were discussed above.

3. Related Figure 3, the difference between the current study and ref. 9 should be discussed.

After completion of the RING1B ChIPs, our data no longer conflicts with Stanton et al. because they only looked at RING1B and not H2AK119ub. We thank the reviewer again for asking us to

look at the actual PRC proteins. We did not see the direct correlation between RING1B and the mark it writes H2AK119ub as expected. The RING1B results are consistent with the Stanton study. We have added the following section to the Discussion section: “Additionally, we found a discrepancy between the PRC1 protein (RING1B) and the PRC1-associated mark H2AK119ub removal, where the actual PRC1 proteins are evicted much more efficiently. This could be a technical artifact behind H2AK119ub ChIPs or could be biological, where cBAF complexes physically interact with cPRC1 to evict the protein complex^{3,7}, but the histone mark remains more stable even without PRC1 present at the locus. An increase in PRC1 binding chromatin in BRG1 ATPase conditional knockout or mutant cells is consistent with our data showing cBAF complexes efficiently evict RING1B proteins, but the loss of H2AK119ub marks in BRG1 ATPase mutants was not explored in this study, so we are unsure whether the difference we see between RING1B eviction and H2AK119ub is universal.³

4. In Discussion, (1) Co-recruitment of an enzyme or enzymes that remove the marks, leaving the rest of the nucleosomes intact, an article Takano, J. et al. PCGF1-PRC1 links chromatin repression with DNA replication during hematopoietic cell lineage commitment. *Nat Commun* 13, 7159 (2022). <https://doi.org/10.1038/s41467-022-34856-8> could be supportive as in this paper the authors revealed SWI/SNF complexes compete with PRC1/2 to evict H3K27me3 without involving the recruitment of demethylase in hematopoietic progenitor cells.

We thank the reviewer for suggesting this excellent study for citation and discussion in our manuscript. It is consistent with our findings and suggests that this phenomenon is not unique to stem cell biology.

To the Discussion Section we added:

Others have shown that BAF competes with PRC complexes to evict H3K27me3 without the co-recruitment of demethylases in a hematopoietic stem cell system, consistent with model 3²⁵.

1. Kadoch C, Williams RT, Calarco JP, Miller EL, Weber CM, Braun SMG, Pulice JL, Chory EJ, Crabtree GR. Dynamics of BAF-Polycomb complex opposition on heterochromatin in normal and oncogenic states. *Nat Genet.* 2017;49(2):213-222.
2. Braun SMG, Kirkland JG, Chory EJ, Husmann D, Calarco JP, Crabtree GR. Rapid and reversible epigenome editing by endogenous chromatin regulators. *Nat Commun.* 2017;8(1):560.
3. Stanton BZ, Hodges C, Calarco JP, Braun SMG, Ku WL, Kadoch C, Zhao K, Crabtree GR. Smarca4 ATPase mutations disrupt direct eviction of PRC1 from chromatin. *Nat Genet.* 2017;49(2):282-288.
4. Fursova NA, Blackledge NP, Nakayama M, Ito S, Koseki Y, Farcas AM, King HW, Koseki H, Klose RJ. Synergy between variant PRC1 complexes defines Polycomb-mediated gene repression. *Mol Cell.* 2019;74(5):1020-1036.e8.
5. Miller EL, Hargreaves DC, Kadoch C, Chang CY, Calarco JP, Hodges C, Buenrostro JD, Cui K, Greenleaf WJ, Zhao K, Crabtree GR. TOP2 synergizes with BAF chromatin remodeling for both resolution and formation of facultative heterochromatin. *Nat Struct Mol Biol.* 2017;24(4):344-352.

6. Hathaway NA, Bell O, Hodges C, Miller EL, Neel DS, Crabtree GR. Dynamics and memory of heterochromatin in living cells. *Cell*. 2012;149(7):1447-1460.
7. Boulay G, Cironi L, Garcia SP, Rengarajan S, Xing YH, Lee L, Awad ME, Naigles B, Iyer S, Broye LC, Keskin T, Cauderay A, Fusco C, Letovanec I, Chebib I, Nielsen PG, Tercier S, Cherix S, Nguyen-Ngoc T, Cote G, Choy E, Provero P, Suvà ML, Rivera MN, Stamenkovic I, Riggi N. The chromatin landscape of primary synovial sarcoma organoids is linked to specific epigenetic mechanisms and dependencies. *Life Sci Alliance*. 2021;4(2):e202000808.

August 15, 2024

RE: Life Science Alliance Manuscript #LSA-2024-02715-TR

Dr. Jacob G Kirkland
Oklahoma Medical Research Foundation
Cell Cycle and Cancer Biology
825 N.E. 13th Street
MS-48
Oklahoma City, OK 73104

Dear Dr. Kirkland,

Thank you for submitting your revised manuscript entitled "Differential Modulation of Polycomb-Associated Histone Marks by cBAF, pBAF, and gBAF Complexes". We would be happy to publish your paper in Life Science Alliance pending final revisions necessary to meet our formatting guidelines.

- please be sure that the authorship listing and order is correct
- please upload your main manuscript text as an editable doc file
- please upload your Tables in editable .doc or excel format; -Tables should be numbered consecutively with Arabic numerals (1, 2, 3, 4); They can be included at the bottom of the main manuscript file or be sent as separate files.
- please upload all figure files as individual ones, including the supplementary figure files; all figure legends should only appear in the main manuscript file
- please use the [10 author names, et al.] format in your references (i.e. limit the author names to the first 10)
- please add callouts for Figure S5 C & F and Figure S6C to your main manuscript

Figure Check:

- please add sizes next to all blots
- Figure S1 and S6E should be uploaded as Source Data instead. Then please renumber remaining Supplemental Figures and update their callouts.

LSA now encourages authors to provide a 30-60 second video where the study is briefly explained. We will use these videos on social media to promote the published paper and the presenting author (for examples, see <https://docs.google.com/document/d/1-UWCfbE4pGcDdcgzcmiuJl2XMBJnxKYeqRvLLrLSo8s/edit?usp=sharing>). Corresponding or first-authors are welcome to submit the video. Please submit only one video per manuscript. The video can be emailed to contact@life-science-alliance.org

A. FINAL FILES:

B. MANUSCRIPT ORGANIZATION AND FORMATTING:

Sincerely,

August 21, 2024

RE: Life Science Alliance Manuscript #LSA-2024-02715-TRR

Dr. Jacob G Kirkland
Oklahoma Medical Research Foundation
Cell Cycle and Cancer Biology
825 N.E. 13th Street
MS-48
Oklahoma City, OK 73104

Dear Dr. Kirkland,

Thank you for submitting your Research Article entitled "Differential Modulation of Polycomb-Associated Histone Marks by cBAF, pBAF, and gBAF Complexes". It is a pleasure to let you know that your manuscript is now accepted for publication in Life Science Alliance. Congratulations on this interesting work.

DISTRIBUTION OF MATERIALS:

Again, congratulations on a very nice paper. I hope you found the review process to be constructive and are pleased with how the manuscript was handled editorially. We look forward to future exciting submissions from your lab.

Sincerely,
